# Big Bird: Transformers for Longer Sequences

**Manzil Zaheer,**     **Guru Guruganesh,**     **Avinava Dubey,**
**Joshua Ainslie,**    **Chris Alberti,**    **Santiago Ontanon,**    **Philip Pham,**
**Anirudh Ravula,**    **Qifan Wang,**    **Li Yang,**    **Amr Ahmed**
Google Research
{manzilz, gurug, avinavadubey}@google.com

## Abstract

Transformers-based models, such as BERT, have been one of the most successful deep learning models for NLP. Unfortunately, one of their core limitations is the quadratic dependency (mainly in terms of memory) on the sequence length due to their full attention mechanism. To remedy this, we propose, BIGBIRD, a sparse attention mechanism that reduces this quadratic dependency to linear. We show that BIGBIRD is a universal approximator of sequence functions and is Turing complete, thereby preserving these properties of the quadratic, full attention model. Along the way, our theoretical analysis reveals some of the benefits of having $O(1)$ global tokens (such as CLS), that attend to the entire sequence as part of the sparse attention mechanism. The proposed sparse attention can handle sequences of length up to 8x of what was previously possible using similar hardware. As a consequence of the capability to handle longer context, BIGBIRD drastically improves performance on various NLP tasks such as question answering and summarization. We also propose novel applications to genomics data.

## 1  Introduction

Models based on Transformers [92], such as BERT [22, 63], are wildly successful for a wide variety of Natural Language Processing (NLP) tasks and consequently are mainstay of modern NLP research. Their versatility and robustness are the primary drivers behind the wide-scale adoption of Transformers. The model is easily adapted for a diverse range of sequence based tasks – as a seq2seq model for translation [92], summarization [66], generation [15], etc. or as a standalone encoders for sentiment analysis [84], POS tagging [65], machine reading comprehension [94], etc. – and it is known to vastly outperform previous sequence models like LSTM [37]. The key innovation in Transformers is the introduction of a self-attention mechanism, which can be evaluated in parallel for each token of the input sequence, eliminating the sequential dependency in recurrent neural networks, like LSTM. This parallelism enables Transformers to leverage the full power of modern SIMD hardware accelerators like GPUs/TPUs, thereby facilitating training of NLP models on datasets of unprecedented size. This ability to train on large scale data has led to surfacing of models like BERT [22] and T5 [75], which pretrain transformers on large general purpose corpora and transfer the knowledge to down-stream task. The pretraining has led to significant improvement in low data regime downstream tasks [51] as well as tasks with sufficient data [102] and thus have been a major force behind the ubiquity of transformers in contemporary NLP.

The self-attention mechanism overcomes constraints of RNNs (namely the sequential nature of RNN) by allowing each token in the input sequence to attend independently to every other token in the sequence. This design choice has several interesting repercussions. In particular, the full self-attention have computational and memory requirement that is quadratic in the sequence length. We note that while the corpus can be large, the sequence length, which provides the context in many applications is very limited. Using commonly available current hardware and model sizes, this requirement translates to roughly being able to handle input sequences of length 512 tokens. This reduces its direct applicability to tasks that require larger context, like QA [60], document classification, etc.

However, while we know that self-attention and Transformers are useful, our theoretical understanding is rudimentary. What aspects of the self-attention model are necessary for its performance? What can we say about the expressivity of Transformers and similar models? Apriori, it was not even clear from the design if the proposed self-attention mechanism was as effective as RNNs. For example, the self-attention does not even obey sequence order as it is permutation equivariant. This concern has been partially resolved, as Yun et al. [105] showed that transformers are expressive enough to capture all continuous sequence to sequence functions with a compact domain. Meanwhile, Pérez et al. [72] showed that the full transformer is Turing Complete (i.e. can simulate a full Turing machine). Two natural questions arise: Can we achieve the empirical benefits of a fully quadratic self-attention scheme using fewer inner-products? Do these sparse attention mechanisms preserve the expressivity and flexibility of the original network?

In this paper, we address both the above questions and produce a sparse attention mechanism that improves performance on a multitude of tasks that require long contexts. We systematically develop BIGBIRD, an attention mechanism whose complexity is linear in the number of tokens (Sec. 2). We take inspiration from graph sparsification methods and understand where the proof for expressiveness of Transformers breaks down when full-attention is relaxed to form the proposed attention pattern. This understanding helped us develop BIGBIRD, which is theoretically as expressive and also empirically useful. In particular, our BIGBIRD consists of three main part:

- A set of $g$ global tokens attending on all parts of the sequence.
- All tokens attending to a set of $w$ local neighboring tokens.
- All tokens attending to a set of $r$ random tokens.

This leads to a high performing attention mechanism scaling to much longer sequence lengths (8x). To summarize, our main **contributions** are:

1. BIGBIRD satisfies all the known theoretical properties of full transformer (Sec. 3). In particular, we show that adding extra tokens allows one to express all continuous sequence to sequence functions with only $O(n)$-inner products. Furthermore, we show that under standard assumptions regarding precision, BIGBIRD is Turing complete.
2. Empirically, we show that the extended context modelled by BIGBIRD benefits variety of NLP tasks. We achieve *state of the art* results for question answering and document summarization on a number of different datasets. Summary of these results are presented in Sec. 4.
3. Lastly, we introduce a novel application of attention based models where long contexts are beneficial: extracting contextual representations of genomics sequences like DNA. With longer masked LM pretraining, BIGBIRD improves performance on downstream tasks such as promoter-region and chromatin profile prediction (Sec. 5).

## 1.1 Related Work

There have been a number of interesting attempts, that were aimed at alleviating the quadratic dependency of Transformers, which can broadly categorized into two directions. First line of work embraces the length limitation and develops method around it. Simplest methods in this category just employ sliding window [94], but in general most work fits in the following general paradigm: using some other mechanism select a smaller subset of relevant contexts to feed in the transformer and optionally iterate, i.e. call transformer block multiple time with different contexts each time. Most prominently, SpanBERT [42], ORQA [54], REALM [34], RAG [57] have achieved strong performance for different tasks. However, it is worth noting that these methods often require significant engineering efforts (like back prop through large scale nearest neighbor search) and are hard to train.

Second line of work questions if full attention is essential and have tried to come up with approaches that do not require full attention, thereby reducing the memory and computation requirements. Prominently, Dai et al. [21], Sukhbaatar et al. [83], Rae et al. [74] have proposed auto-regressive models that work well for left-to-right language modeling but suffer in tasks which require bidirectional context. Child et al. [16] proposed a sparse model that reduces the complexity to $O(n\sqrt{n})$, Kitaev et al. [49] further reduced the complexity to $O(n \log(n))$ by using LSH to compute nearest neighbors. Ye et al. [104] proposed binary partitions of the data where as Qiu et al. [73] reduced complexity by using block sparsity. Recently, Longformer [8] introduced a localized sliding window based mask with few global mask to reduce computation and extended BERT to longer sequence based tasks. Finally, our work is closely related to and built on the work of Extended Transformers Construction [4]. This work was designed to encode structure in text for transformers. The idea of global tokens was used extensively by them to achieve their goals. Our theoretical work can be seen as providing a justification for the success of these models as well. It is important to note that most of the

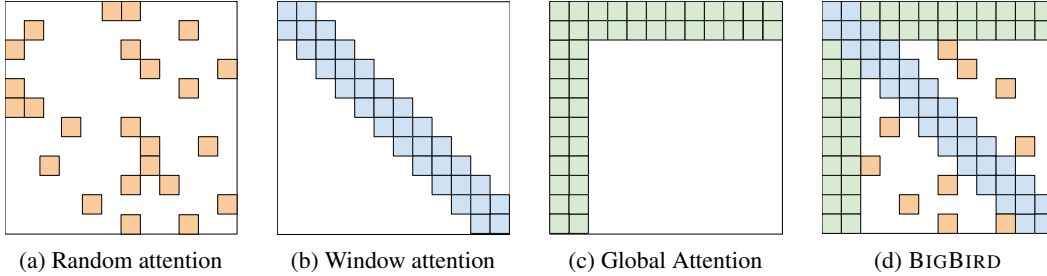

| (a) Random attention | (b) Window attention | (c) Global Attention | (d) BIGBIRD |

Figure 1: Building blocks of the attention mechanism used in BIGBIRD. White color indicates absence of attention. (a) random attention with $r = 2$, (b) sliding window attention with $w = 3$ (c) global attention with $g = 2$. (d) the combined BIGBIRD model.

aforementioned methods are heuristic based and empirically are not as versatile and robust as the original transformer, i.e. the same architecture do not attain SoTA on multiple standard benchmarks. (There is one exception of Longformer which we include in all our comparisons, see App. E.3 for a more detailed comparison). Moreover, these approximations do not come with theoretical guarantees.

## 2  BIGBIRD Architecture

In this section, we describe the BIGBIRD model using the *generalised attention mechanism* that is used in each layer of transformer operating on an input sequence $\boldsymbol{X} = (\boldsymbol{x}_1, ..., \boldsymbol{x}_n) \in \mathbb{R}^{n \times d}$. The *generalized attention mechanism* is described by a directed graph $D$ whose vertex set is $[n] = \{1, \ldots, n\}$. The set of arcs (directed edges) represent the set of inner products that the attention mechanism will consider. Let $N(i)$ denote the out-neighbors set of node $i$ in $D$, then the $i^{\text{th}}$ output vector of the generalized attention mechanism is defined as

$$\text{ATTN}_D(\boldsymbol{X})_i = \boldsymbol{x}_i + \sum_{h=1}^{H} \sigma \left( Q_h(\boldsymbol{x}_i) K_h(\boldsymbol{X}_{N(i)})^T \right) \cdot V_h(\boldsymbol{X}_{N(i)}) \tag{AT}$$

where $Q_h, K_h : \mathbb{R}^d \to \mathbb{R}^m$ are query and key functions respectively, $V_h : \mathbb{R}^d \to \mathbb{R}^d$ is a value function, $\sigma$ is a scoring function (e.g. softmax or hardmax) and $H$ denotes the number of heads. Also note $X_{N(i)}$ corresponds to the matrix formed by only stacking $\{\boldsymbol{x}_j : j \in N(i)\}$ and not all the inputs.

If $D$ is the complete digraph, we recover the full quadratic attention mechanism of Vaswani et al. [92]. To simplify our exposition, we will operate on the adjacency matrix $A$ of the graph $D$ even though the underlying graph maybe sparse. To elaborate, $A \in [0,1]^{n \times n}$ with $A(i,j) = 1$ if query $i$ attends to key $j$ and is zero otherwise. For example, when $A$ is the ones matrix (as in BERT), it leads to quadratic complexity, since all tokens attend on every other token. This view of self-attention as a fully connected graph allows us to exploit existing graph theory to help reduce its complexity. The problem of reducing the quadratic complexity of self-attention can now be seen as a *graph sparsification problem*. It is well-known that random graphs are expanders and can approximate complete graphs in a number of different contexts including in their spectral properties [80, 38]. We believe sparse random graph for attention mechanism should have two desiderata: small average path length between nodes and a notion of locality, each of which we discuss below.

Let us consider the simplest random graph construction, known as Erdős-Rényi model, where each edge is independently chosen with a fixed probability. In such a random graph with just $\tilde{\Theta}(n)$ edges, the shortest path between any two nodes is logarithmic in the number of nodes [17, 43]. As a consequence, such a random graph approximates the complete graph spectrally and its second eigenvalue (of the adjacency matrix) is quite far from the first eigenvalue [9, 10, 6]. This property leads to a rapid mixing time for random walks in the grpah, which informally suggests that information can flow fast between any pair of nodes. Thus, we propose a sparse attention where each query attends over $r$ random number of keys i.e. $A(i,\cdot) = 1$ for $r$ randomly chosen keys (see Fig. 1a).

The second viewpoint which inspired the creation of BIGBIRD is that most contexts within NLP and computational biology have data which displays a great deal of *locality of reference*. In this phenomenon, a great deal of information about a token can be derived from its neighboring tokens. Most pertinently, Clark et al. [19] investigated self-attention models in NLP tasks and concluded that that neighboring inner-products are extremely important. The concept of locality, proximity of tokens in linguistic structure, also forms the basis of various linguistic theories such as transformational-generative grammar. In the terminology of graph theory, clustering coefficient is a measure of locality

of connectivity, and is high when the graph contains many cliques or near-cliques (subgraphs that are almost fully interconnected). Simple Erdős-Rényi random graphs do not have a high clustering coefficient [85], but a class of random graphs, known as small world graphs, exhibit high clustering coefficient [95]. A particular model introduced by Watts and Strogatz [95] is of high relevance to us as it achieves a good balance between average shortest path and the notion of locality. The generative process of their model is as follows: Construct a regular ring lattice, a graph with $n$ nodes each connected to $w$ neighbors, $w/2$ on each side.

In other words we begin with a sliding window on the nodes. Then a random subset ($k\%$) of all connections is replaced with a random connection. The other ($100 - k$)% local connections are retained. However, deleting such random edges might be inefficient on modern hardware, so we retain it, which will not affect its properties. In summary, to capture

Table 1: Building block comparison @512

| Model | MLM | SQuAD | MNLI |
|---|---|---|---|
| BERT-base | 64.2 | 88.5 | 83.4 |
| Random (R) | 60.1 | 83.0 | 80.2 |
| Window (W) | 58.3 | 76.4 | 73.1 |
| R + W | 62.7 | 85.1 | 80.5 |

these local structures in the context, in BIGBIRD, we define a sliding window attention, so that during self attention of width $w$, query at location $i$ attends from $i - w/2$ to $i + w/2$ keys. In our notation, $A(i, i - w/2 : i + w/2) = 1$ (see Fig. 1b). As an initial sanity check, we performed basic experiments to test whether these intuitions are sufficient in getting performance close to BERT like models, while keeping attention linear in the number of tokens. We found that random blocks and local window were insufficient in capturing all the context necessary to compete with the performance of BERT.

The final piece of BIGBIRD is inspired from our theoretical analysis (Sec. 3), which is critical for empirical performance. More specifically, our theory utilizes the importance of "global tokens" (tokens that attend to all tokens in the sequence and to whom all tokens attend to (see Fig. 1c). These global tokens can be defined in two ways:

- BIGBIRD-ITC: In internal transformer construction (ITC), we make some existing tokens "global", which attend over the entire sequence. Concretely, we choose a subset $G$ of indices (with $g := |G|$), such that $A(i, :) = 1$ and $A(:, i) = 1$ for all $i \in G$.
- BIGBIRD-ETC: In extended transformer construction (ETC), we include additional "global" tokens such as CLS. Concretely, we add $g$ global tokens that attend to all existing tokens. In our notation, this corresponds to creating a new matrix $B \in [0, 1]^{(N+g) \times (N+g)}$ by adding $g$ rows to matrix $A$, such that $B(i, :) = 1$, and $B(:, i) = 1$ for all $i \in \{1, 2, \dots g\}$, and $B(g + i, g + j) = A(i, j) \forall i, j \in \{1, \dots, N\}$. This adds extra location to store context and as we will see in the experiments improves performance.

The final attention mechanism for BIGBIRD (Fig. 1d) has all three of these properties: queries attend to $r$ random keys, each query attends to $w/2$ tokens to the left of its location and $w/2$ to the right of its location and they contain $g$ global tokens (The global tokens can be from existing tokens or extra added tokens). We provide implementation details in App. D.

## 3 Theoretical Results about Sparse Attention Mechanism

In this section, we will show that that sparse attention mechanisms are as powerful and expressive as full-attention mechanisms in two respects. First, we show that when sparse attention mechanisms are used in a standalone encoder (such as BERT), they are Universal Approximators of sequence to sequence functions in the style of Yun et al. [105]. We note that this property was also explored theoretically in contemporary work Yun et al. [106]. Second, unlike [106], we further show that sparse encoder-decoder transformers are Turing Complete (assuming the same conditions defined in [72]). Complementing the above positive results, we also show that moving to a sparse-attention mechanism incurs a cost, i.e. there is no free lunch. In Sec. 3.4, we show lower bounds by exhibiting a natural task where any sufficiently sparse mechanism will require polynomially more layers.

### 3.1 Notation

The complete Transformer *encoder* stack is nothing but the repeated application of a single-layer encoder (with independent parameters). We denote class of such Transformer encoders stack, defined using generalized encoder (Sec. 2), by $\mathcal{T}_D^{H,m,q}$ which consists of $H$-heads with head size $m$ and $q$ is the hidden layer size of the output network, and the attention layer is defined by the directed graph $D$.

The key difference between our proposed attention mechanism to that of Vaswani et al. [92], Yun et al. [105] is that we add a special token at the beginning of each sequence and assign it a special vector.

We will refer to this as $\boldsymbol{x}_0$. Therefore our graph $D$ will have vertex set $\{0\} \cup [n] = \{0, 1, 2, \ldots, n\}$. We will assume that this extra node and its respective vector will be dropped at the final output layer of transformer. To avoid cumbersome notation, we will still treat transformer as mapping sequences $\boldsymbol{X} \in \mathbb{R}^{n \times d}$ to $\mathbb{R}^{n \times d}$. We will also allow the transformer to append position embeddings $E \in \mathbb{R}^{d \times n}$ to matrix $X$ in the input layer.

Finally, we need to define the function class and distance measure for proving universal approximation property. Let $\mathcal{F}_{CD}$ denote the set of continuous functions $f : [0, 1]^{n \times d} \to \mathbb{R}^{n \times d}$ which are continuous with respect to the topology defined by $\ell_p$ norm. Recall for any $p \geq 1$, the $\ell_p$ distance is $d_p(f_1, f_2) = \left( \int \|f_1(X) - f_2(X)\|_p^p dX \right)^{1/p}$.

## 3.2 Universal Approximators

**Definition 1.** *The star-graph $S$ centered at $0$ is the graph defined on $\{0, \ldots, n\}$. The neighborhood of all vertices $i$ is $N(i) = \{0, i\}$ for $i \in \{1 \ldots n\}$ and $N(0) = \{1, \ldots n\}$.*

Our main theorem is that the sparse attention mechanism defined by any graph containing $S$ is a universal approximator:

**Theorem 1.** *Given $1 < p < \infty$ and $\epsilon > 0$, for any $f \in \mathcal{F}_{CD}$, there exists a transformer with sparse-attention, $g \in \mathcal{T}_D^{H,m,q}$ such that $d_p(f, g) \leq \epsilon$ where $D$ is any graph containing star graph $S$.*

To prove the theorem, we will follow the standard proof structure outlined in [105].

**Step 1: Approximate $\mathcal{F}_{CD}$ by piece-wise constant functions.** Since $f$ is a continuous function with bounded domain $[0, 1)^{n \times d}$, we will approximate it with a suitable piece-wise constant function. This is accomplished by a suitable partition of the region $[0, 1)$ into a grid of granularity $\delta$ to get a discrete set $\mathbb{G}_\delta$. Therefore, we can assume that we are dealing with a function $\bar{f} : \mathbb{G}_\delta \to \mathbb{R}^{n \times d}$, where $d_p(f, \bar{f}) \leq \frac{\epsilon}{3}$.

**Step 2: Approximate piece-wise constant functions by modified transformers.** This is the key step of the proof where the self-attention mechanism is used to generate a *contextual-mapping* of the input. Informally, a contextual mapping is a unique code for the pair consisting of a matrix $(\boldsymbol{X}, \boldsymbol{x}_i)$ and a column. Its uniqueness allows the Feed forward layers to use each code to map it to a unique output column.

The main technical challenge is computing the contextual mapping using only sparse attention mechanism. This was done in [105] using a "selective" shift operator which shift up entries that are in a specific interval. Key to their proof was the fact that the shift, was exactly the range of the largest entry to the smallest entry.

Creating a contextual mapping with a sparse attention mechanism is quite a challenge. In particular, because each query only attends to a few keys, it is not at all clear that sufficient information can be corralled to make a contextual embedding of the entire matrix. To get around this, we develop a sparse shift operator which shifts the entries of the matrices if they lie in a certain range. The exact amount of the shift is controlled by the directed sparse attention graphg $D$. The second key ingredient is the use of additional global token. By carefully applying the operator to a set of chosen ranges, we will show that each column will contain a unique mapping of the full mapping. Therefore, we can augment the loss of inner-products in the self attention mechanism by using multiple layers and an auxiliary global token.

**Step 3: Approximate modified transformers by original Transformers**: The final step is to approximate the modified transformers by the original transformer which uses ReLU and softmax.

We provide the full details in App. A.

## 3.3 Turing Completeness

Transformers are a very general class. In the original paper of Vaswani et al. [92], they were used in both an encoder and a decoder. While the previous section outlined how powerful just the encoders were, another natural question is to ask what the additional power of both a decoder along with an encoder is? Pérez et al. [72] showed that the full transformer based on a quadratic attention mechanism is Turing Complete. This result makes one unrealistic assumption, which is that the model works on arbitrary precision model. Of course, this is necessary as otherwise, Transformers are bounded finite state machines and cannot be Turing Complete.

It is natural to ask if the full attention mechanism is necessary. Or can a sparse attention mechanism also be used to simulate any Turing Machine? We show that this is indeed the case: we can use a sparse encoder and sparse decoder to simulate any Turing Machine.

To use the sparse attention mechanism in the transformer architecture, we need to define a suitable modification where each token only reacts to previous tokens. Unlike the case for BERT, where the entire attention mechanism is applied once, in full transformers, the sparse attention mechanism at decoder side is used token by token. Secondly the work of Pérez et al. [72], uses each token as a representation of the tape history and uses the full attention to move and retrieve the correct tape symbol. Most of the construction of Pérez et al. [72] goes through for sparse attentions, except for their addressing scheme to point back in history (Lemma B.4 in [72]). We show how to simulate this using a sparse attention mechanism and defer the details to App. B.

### 3.4 Limitations

We demonstrate a natural task which can be solved by the full attention mechanism in $O(1)$-layers. However, under standard complexity theoretic assumptions, this problem requires $\tilde{\Omega}(n)$-layers for any sparse attention layers with $\tilde{O}(n)$ edges (not just BIGBIRD). (Here $\tilde{O}$ hides poly-logarthmic factors). Consider the simple problem of finding the corresponding furthest vector for each vector in the given sequence of length $n$. Formally,

**Task 1.** Given $n$ unit vectors $\{u_1, \ldots, u_n\}$, find $f(u_1, \ldots, u_n) \rightarrow (u_{1^*}, \ldots, u_{n^*})$ where for a fixed $j \in [n]$, we define $j^* = \arg\max_k \|u_k - u_j\|_2^2$.

Finding vectors that are furthest apart boils down to minimize inner product search in case of unit vectors. For a full-attention mechanism with appropriate query and keys, this task is very easy as we can evaluate all pair-wise inner products.

The impossibility for sparse-attention follows from hardness results stemming from Orthogonal Vector Conjecture(OVC) [1, 2, 7, 97]. The OVC is a widely used assumption in fine-grained complexity. Informally, it states that one cannot determine if the minimum inner product among $n$ boolean vectors is 0 in subquadratic time. In App. C, we show a reduction using OVC to show that if a transformer $g \in \mathcal{T}_D^{H=1,m=2d,q=0}$ for any sparse directed graph $D$ can evaluate the Task 1, it can solve the orthogonal vector problem.

**Proposition 1.** *There exists a single layer full self-attention $g \in \mathcal{T}^{H=1,m=2d,q=0}$ that can evaluate Task 1, i.e. $g(u_1, ..., u_n) = [u_{1^*}, \ldots, u_{n^*}]$, but for any sparse-attention graph $D$ with $\tilde{O}(n)$ edges (i.e. inner product evaluations), would require $\tilde{\Omega}(n^{1-o(1)})$ layers.*

We give a formal proof of this fact in App. C.

## 4 Experiments: Natural Language Processing

In this section our goal is to showcase benefits of modeling longer input sequence for NLP tasks, for which we select three representative tasks. We begin with basic masked language modeling (MLM; Devlin et al. [22]) to check if better contextual representations can be learnt by utilizing longer contiguous sequences. Next, we consider QA with supporting evidence, for which capability to handle longer sequence would allow us to retrieve more evidence using crude systems like TF-IDF/BM25. Finally, we tackle long document classification where discriminating information may not be located in first 512 tokens. Below we summarize the results for BIGBIRD using sequence length 4096[1], while we defer all other setup details including computational resources, batch size, step size, to App. E.

**Pretraining and MLM**   We follow [22, 63] to create base and large versions of BIGBIRD and pretrain it using MLM objective. This task involves predicting a random subset of tokens which have been masked out. We use four standard data-sets for pretraining (listed in App. E.1, Tab. 9), warm-starting from the public RoBERTa checkpoint[2]. We compare performance in predicting the masked out tokens in terms of bits per character, following [8]. As seen in App. E.1, Tab. 10, both BIGBIRD and Longformer perform better than limited length RoBERTa, with BIGBIRD-ETC performing the best. We note that we trained our models on a reasonable $16GB$ memory/chip with batch size of 32-64. Our memory efficiency is due to efficient blocking and sparsity structure of the sparse attention mechanism described in Sec. 2.

Table 2: QA Dev results using Base size models. We report accuracy for WikiHop and F1 for HotpotQA, Natural Questions, and TriviaQA.

| Model | HotpotQA | | | NaturalQ | | TriviaQA | WikiHop |
|---|---|---|---|---|---|---|---|
| | Ans | Sup | Joint | LA | SA | Full | MCQ |
| RoBERTa | 73.5 | 83.4 | 63.5 | - | - | 74.3 | 72.4 |
| Longformer | 74.3 | 84.4 | 64.4 | - | - | 75.2 | 75.0 |
| BIGBIRD-ITC | **75.7** | 86.8 | 67.7 | 70.8 | 53.3 | **79.5** | **75.9** |
| BIGBIRD-ETC | 75.5 | **87.1** | **67.8** | **73.9** | **54.9** | 78.7 | **75.9** |

**Question Answering (QA)**  We considered following four challenging datasets:

1. Natural Questions [52]: For the given question, find a short span of answer (SA) from the given evidences as well highlight the paragraph from the given evidences containing information about the correct answer (LA).
2. HotpotQA-distractor [101]: Similar to natural questions, it requires finding the answer (Ans) as well as the supporting facts (Sup) over different documents needed for multi-hop reasoning from the given evidences.
3. TriviaQA-wiki [41]: We need to provide an answer for the given question using provided Wikipedia evidence, however, the answer might not be present in the given evidence. On a smaller *verified* subset of question, the given evidence is guaranteed to contain the answer. Nevertheless, we model the answer as span selection problem in this case as well.
4. WikiHop [96]: Chose correct option from multiple-choice questions (MCQ), by aggregating information spread across multiple documents given in the evidences.

As these tasks are very competitive, multiple highly engineered systems have been designed specific each dataset confirming to respective output formats. For a fair comparison, we had to use some additional regularization for training BIGBIRD, details of which are provided in App. E.2 along with exact architecture description. We experiment using the base sized model and select the best configuration on the development set for each dataset (as reported in Tab. 2). We can see that BIGBIRD-ETC, with expanded global tokens consistently outperforms all other models. Thus, we chose this configuration to train a large sized model to be used for evaluation on the hidden test set.

In Tab. 3, we compare BIGBIRD-ETC model to top-3 entries from the leaderboard excluding BIGBIRD. One can clearly see the importance of using longer context as both Longformer and BIGBIRD outperform models with smaller contexts. Also, it is worth noting that BIGBIRD submission is a single model, whereas the other top-3 entries for Natural Questions are ensembles, which might explain the slightly lower accuracy in exact answer phrase selection.

**Classification**  We experiment on datasets of different lengths and contents, specifically various document classification and GLUE tasks. Following BERT, we used one layer with cross entropy loss on top of the first [CLS] token. We see that gains of using BIGBIRD are more significant when we have longer documents and fewer training examples. For instance, using base sized model,

Table 3: Fine-tuning results on **Test** set for QA tasks. The Test results (F1 for HotpotQA, Natural Questions, TriviaQA, and Accuracy for WikiHop) have been picked from their respective leaderboard. For each task the top-3 leaders were picked not including BIGBIRD-etc. **For Natural Questions Long Answer (LA), TriviaQA, and WikiHop, BIGBIRD-ETC is the new state-of-the-art**. On HotpotQA we are third in the leaderboard by F1 and second by Exact Match (EM).

| Model | HotpotQA | | | NaturalQ | | TriviaQA | | WikiHop |
|---|---|---|---|---|---|---|---|---|
| | Ans | Sup | Joint | LA | SA | Full | Verified | MCQ |
| HGN [26] | **82.2** | 88.5 | **74.2** | - | - | - | - | - |
| GSAN | 81.6 | 88.7 | 73.9 | - | - | - | - | - |
| ReflectionNet [32] | - | - | - | 77.1 | **64.1** | - | - | - |
| RikiNet-v2 [61] | - | - | - | 76.1 | 61.3 | - | - | - |
| Fusion-in-Decoder [39] | - | - | - | - | - | 84.4 | 90.3 | - |
| SpanBERT [42] | - | - | - | - | - | 79.1 | 86.6 | - |
| MRC-GCN [88] | - | - | - | - | - | - | - | 78.3 |
| MultiHop [14] | - | - | - | - | - | - | - | 76.5 |
| Longformer [8] | 81.2 | 88.3 | 73.2 | - | - | 77.3 | 85.3 | 81.9 |
| BIGBIRD-ETC | 81.2 | **89.1** | 73.6 | **77.8** | 57.9 | **84.5** | **92.4** | **82.3** |

Table 4: Summarization ROUGE score for long documents.

| Model | | Arxiv | | | PubMed | | | BigPatent | | |
|---|---|---|---|---|---|---|---|---|---|---|
| | | R-1 | R-2 | R-L | R-1 | R-2 | R-L | R-1 | R-2 | R-L |
| Prior Art | SumBasic [68] | 29.47 | 6.95 | 26.30 | 37.15 | 11.36 | 33.43 | 27.44 | 7.08 | 23.66 |
| | LexRank [25] | 33.85 | 10.73 | 28.99 | 39.19 | 13.89 | 34.59 | 35.57 | 10.47 | 29.03 |
| | LSA [98] | 29.91 | 7.42 | 25.67 | 33.89 | 9.93 | 29.70 | - | - | - |
| | Attn-Seq2Seq [86] | 29.30 | 6.00 | 25.56 | 31.55 | 8.52 | 27.38 | 28.74 | 7.87 | 24.66 |
| | Pntr-Gen-Seq2Seq [77] | 32.06 | 9.04 | 25.16 | 35.86 | 10.22 | 29.69 | 33.14 | 11.63 | 28.55 |
| | Long-Doc-Seq2Seq [20] | 35.80 | 11.05 | 31.80 | 38.93 | 15.37 | 35.21 | - | - | - |
| | Sent-CLF [82] | 34.01 | 8.71 | 30.41 | 45.01 | 19.91 | 41.16 | 36.20 | 10.99 | 31.83 |
| | Sent-PTR [82] | 42.32 | 15.63 | 38.06 | 43.30 | 17.92 | 39.47 | 34.21 | 10.78 | 30.07 |
| | Extr-Abst-TLM [82] | 41.62 | 14.69 | 38.03 | 42.13 | 16.27 | 39.21 | 38.65 | 12.31 | 34.09 |
| | Dancer [31] | 42.70 | 16.54 | 38.44 | 44.09 | 17.69 | 40.27 | - | - | - |
| Base | Transformer | 28.52 | 6.70 | 25.58 | 31.71 | 8.32 | 29.42 | 39.66 | 20.94 | 31.20 |
| | + RoBERTa [76] | 31.98 | 8.13 | 29.53 | 35.77 | 13.85 | 33.32 | 41.11 | 22.10 | 32.58 |
| | + Pegasus [108] | 34.81 | 10.16 | 30.14 | 39.98 | 15.15 | 35.89 | 43.55 | 20.43 | 31.80 |
| | BIGBIRD-RoBERTa | 41.22 | 16.43 | 36.96 | 43.70 | 19.32 | 39.99 | 55.69 | 37.27 | 45.56 |
| Large | Pegasus (Reported) [108] | 44.21 | 16.95 | 38.83 | 45.97 | 20.15 | 41.34 | 52.29 | 33.08 | 41.75 |
| | Pegasus (Re-eval) | 43.85 | 16.83 | 39.17 | 44.53 | 19.30 | 40.70 | 52.25 | 33.04 | 41.80 |
| | BIGBIRD-Pegasus | **46.63** | **19.02** | **41.77** | **46.32** | **20.65** | **42.33** | **60.64** | **42.46** | **50.01** |

BIGBIRD improves state-of-the-art for Arxiv dataset by about **5**% **points**. On Patents dataset, there is improvement over using simple BERT/RoBERTa, but given the large size of training data the improvement over SoTA (which is not BERT based) is not significant. Note that this performance gain is not seen for much smaller IMDb dataset. Along with experimental setup detail, we present detailed results in App. E.4 which show competitive performance.

## 4.1 Encoder-Decoder Tasks

For an encoder-decoder setup, one can easily see that both suffer from quadratic complexity due to the full self attention. We focus on introducing the sparse attention mechanism of BIGBIRD only at the encoder side. This is because, in practical generative applications, the length of output sequence is typically small as compared to the input. For example for text summarization, we see in realistic scenarios (c.f. App. E.5 Tab. 18) that the median output sequence length is $\sim 200$ where as the input sequence's median length is $> 3000$. For such applications, it is more efficient to use sparse attention mechanism for the encoder and full self-attention for the decoder.

**Summarization** Document summarization is a task of creating a short and accurate summary of a text document. We used three long document datasets for testing our model details of which are mention in Tab. 18. In this paper we focus on abstractive summarization of long documents where using a longer contextual encoder should improve performance. The reasons are two fold: First, the salient content can be evenly distributed in the long document, not just in first 512 tokens, and this is by design in the BigPatents dataset [78]. Second, longer documents exhibit a richer discourse structure and summaries are considerably more abstractive, thereby observing more context helps. As has been pointed out recently [76, 108], pretraining helps in generative tasks, we warm start from our general purpose MLM pretraining on base-sized models as well as utilizing state-of-the-art summarization specific pretraining from Pegasus [108] on large-sized models. The results of training BIGBIRD sparse encoder along with full decoder on these long document datasets are presented in Tab. 4. We can clearly see modeling longer context brings significant improvement. Along with hyperparameters, we also present results on shorter but more widespread datasets in App. E.5, which show that using sparse attention does not hamper performance either.

## 5  Experiments: Genomics

There has been a recent upsurge in using deep learning for genomics data [87, 107, 13], which has resulted in improved performance on several biologically-significant tasks such as promoter site prediction [71], methylation analysis [55], predicting functional effects of non-coding variant [110], etc. These approaches consume DNA sequence fragments as inputs, and therefore we believe longer input sequence handling capability of BIGBIRD would be beneficial as many functional effects

in DNA are highly non-local [12]. Furthermore, taking inspiration from NLP, we learn powerful contextual representations for DNA fragments utilizing abundant unlabeled data (e.g. human reference genome, Saccharomyces Genome Database) via MLM pretraining. Next, we showcase that our long input BigBird along with the proposed pretraining significantly improves performances in two downstream tasks. Detailed experimental setup for the two tasks are provided in App. F.

**Pre-training and MLM**   As explored in Liang [58], instead of operating on base pairs, we propose to first segment DNA into tokens so as to further increase the context length (App. F, Fig. 7). In particular, we build a byte-pair encoding [50] table for the DNA sequence of size 32K, with each token representing 8.78 base pairs on average. We learn contextual representation of these token on the human reference genome (GRCh37)[3] using MLM objective. We then report the bits per character (BPC) on a held-out set in Tab. 5. We find that attention based contextual representation of DNA does improve BPC, which is further improved by using longer context.

Table 5: MLM BPC

| Model | BPC |
|---|---|
| SRILM [58] | 1.57 |
| BERT (sqln. 512) | 1.23 |
| BigBird (sqln. 4096) | **1.12** |

**Promoter Region Prediction**   Promoter is a DNA region typically located upstream of the gene, which is the site of transcription initiation. Multiple methods have been proposed to identify the promoter regions in a given DNA sequence [100, 59, 11, 99, 71], as it is an important first step in understanding gene regulation. The corresponding machine learning task is to classify a given DNA fragment as promoter or non-promoter sequence. We use the dataset compiled by Oubounyt et al. [71] which was built from Eukaryotic Promoter Database (EPDnew) [24] [4]. We finetuned the pretrained BigBird model from above, using the training data and report F1 on test dataset. We compare our results to the previously reported best method in Tab. 6. We see that BigBird achieve nearly perfect accuracy with a 5% jump from the previous best reported accuracy.

Table 6: Comparison.

| Model | F1 |
|---|---|
| CNNProm [91] | 69.7 |
| DeePromoter [71] | 95.6 |
| BigBird | **99.9** |

**Chromatin-Profile Prediction**   Non-coding regions of DNA do not code for proteins. Majority of diseases and other trait associated single-nucleotide polymorphism are correlated to non-coding genomic variations [110, 46]. Thus, understanding the functional effects of non-coding regions of DNA is a very important task. An important step in this process, as defined by Zhou and Troyanskaya [110], is to predict large-scale chromatin-profiling from non-coding genomic sequence. To this effect, DeepSea [110], compiled 919 chromatin-profile of 2.4M non-coding variants from Encyclopedia of DNA Elements (ENCODE)[5] and Roadmap Epigenomics projects[6]. The corresponding ML task is to predict, for a given non-coding region of DNA, these 919 chromatin-profile including 690 transcription factors (TF) binding profiles for 160 different TFs, 125 DNase I sensitivity (DHS) profiles and 104 histone-mark (HM) profiles. We jointly learn 919 binary classifiers to predict these functional effects from sequence of DNA fragments. On held-out chromosomes, we compare AUC with the baselines in Tab. 7 and see that we significantly improve on performance on the harder task HM, which is known to have longer-range correlations [27] than others.

Table 7: Chromatin-Profile Prediction

| Model | TF | HM | DHS |
|---|---|---|---|
| gkm-SVM [30] | 89.6 | - | - |
| DeepSea [110] | 95.8 | 85.6 | **92.3** |
| BigBird | **96.1** | **88.7** | 92.1 |

## 6   Conclusion

We propose BigBird: a sparse attention mechanism that is linear in the number of tokens. BigBird satisfies a number of theoretical results: it is a universal approximator of sequence to sequence functions and is also Turing complete. Theoretically, we use the power of extra global tokens preserve the expressive powers of the model. We complement these results by showing that moving to sparse attention mechanism do incur a cost. Empirically, BigBird gives *state-of-the-art* performance on a number of NLP tasks such as question answering and long document classification. We further introduce attention based contextual language model for DNA and fine-tune it for down stream tasks such as promoter region prediction and predicting effects of non-coding variants.

## Broader Impacts

**Inference Efficiency:** Quadratic attention mechanisms cannot capture long range dependencies which exist in natural text and other datasets. Moreover, there is a growing concern in the ML community about the resource and energy requirement training large scale systems [81]. Moreover, that sparse, computationally efficient systems, like BIGBIRD, can capture long range dependencies in an energy efficient way without losing expressive power.

**Wide Applicability:** Beyond the impact of our model on NLP tasks that require longer context, our proposed contextualized representations of DNA using attention based models, should help in better modeling effects of longer sequences of DNA. Our effort continues a long line of research that bridges the gap between computational models designed for NLP and those for computational biology.

## Footnotes

[1]code available at http://goo.gle/bigbird-transformer

[2]https://github.com/pytorch/fairseq/tree/master/examples/roberta

[3] https://www.ncbi.nlm.nih.gov/assembly/GCF_000001405.13/

[4] https://epd.epfl.ch/human/human_database.php?db=human

[5] https://www.encodeproject.org/

[6] http://www.roadmapepigenomics.org/

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
