[Supplementary Material]

# Big Bird: Transformers for Longer Sequences – Appendix

## A   Universal Approximators

### A.1   Notation

We begin by setting up some notations following Pérez et al. [72] to formally describe the complete architecture of Transformers. A single layer of Transformer encoder is a parametric function Enc receiving a sequence $\boldsymbol{X} = (\boldsymbol{x}_1, ..., \boldsymbol{x}_n)$ of vectors in $\mathbb{R}^d$ and returning a sequence $\boldsymbol{Z} = (\boldsymbol{z}_1, ..., \boldsymbol{z}_n)$ of the same length. Each $\boldsymbol{z}_i$ is a $d$ dimensional vector as well. We interchangeably treat the sequence $\boldsymbol{X}$ as a matrix in $\mathbb{R}^{n \times d}$. Enc has two components:

1. An attention mechanism ATTN that takes in the sequence $\boldsymbol{X}$ and returns sequence $(\boldsymbol{a}_1, ..., \boldsymbol{a}_n)$ of the same length and dimensionality; and

2. A two layer fully connected network $O$ that takes in a vector in $\mathbb{R}^d$ and returns a vector in $\mathbb{R}^d$.

Then $i$-th output vector of $\text{Enc}(\boldsymbol{X})$ is computed as follows:

$$\boldsymbol{z}_i = O(\boldsymbol{a}_i) + \boldsymbol{a}_i \qquad \text{where} \qquad \boldsymbol{a}_i = \text{ATTN}(\boldsymbol{X})_i + \boldsymbol{x}_i \tag{1}$$

Now it remains to define ATTN and $O$ which we do next.

As described in Sec. 2, an attention mechanism is parameterized by three functions: $Q, K, V :$ $\mathbb{R}^d \to \mathbb{R}^m$. In this paper, we assume that they are simply matrix products: $Q(\boldsymbol{x}) = \boldsymbol{x}W_Q$, $K(\boldsymbol{x}) = \boldsymbol{x}W_K$, and $V(\boldsymbol{x}) = \boldsymbol{x}W_V$, where $W_Q, W_K, W_V \in \mathbb{R}^{d \times m}$ and $W_V \in \mathbb{R}^{d \times d}$. In reality a multi-headed attention is used, i.e. we have not only one, but $H$-sets of Query/Key/Value weight matrices, $W_Q^h, W_V^h, W_K^h$ for $h = 1, ..., H$. Thus, for a directed graph $D$ over $[n]$, the $i^{\text{th}}$ output vector of the generalized attention mechanism would be

$$\text{ATTN}_D(\boldsymbol{X})_i = \sum_{h=1}^{H} \sigma \left( (\boldsymbol{x}_i W_Q^h)(\boldsymbol{X}_{N(i)} W_K^h)^T \right) \cdot (\boldsymbol{X}_{N(i)} W_V^h) \tag{AT}$$

where $N(i)$ denote the out-neighbors set of node $i$ in $D$. In other words, the set of arcs (directed edges) in $D$ represents the set of inner products that our attention mechanism will consider. Also recall that $\sigma$ is a scoring function such as softmax or hardmax.

Lastly, we define the output fully connected network as follows:

$$O(\boldsymbol{a}_i) = \text{ReLU}\left(\boldsymbol{a}_i W_1 + b_1\right) W_2 \cdot + b_2 \tag{FF}$$

Here $W_1 \in \mathbb{R}^{d \times q}$, $W_2 \in \mathbb{R}^{q \times d}$, $b_1 \in \mathbb{R}^p$, and $b_2 \in \mathbb{R}^d$ are parameters of output network $O$.

**Additional Notation**   We introduce a few pieces of additional notation that will be useful. Let $[a, b)_\delta = \{a, a + \delta, \ldots, a + \lfloor \frac{b-a}{\delta} \rfloor \cdot \delta\}$. Therefore, $[0, 1)_\delta = \{0, \delta, 2\delta, \ldots, (1 - \delta)\}$. We use $\mathbf{1}[\mathcal{E}]$ to denote the indicator variable; it is 1 if the event $\mathcal{E}$ occurs and 0 otherwise.

### A.2   Proof

In this section, we will present the full proof of theorem 1. The proof will contain three parts. The first and the third part will largely follow standard techniques. The main innovation lies is in the second part.

#### A.2.1   Approximate $\mathcal{F}_{CD}$ by piece-wise constant functions

First, we consider a suitable partition of the region $(0, 1)$ into a grid of granularity $\delta$, which we denote by $G_\delta$. We do this using Lemma 8 from Yun et al. [105], which we restate for completeness:

**Lemma 1** (Lemma 8 [105]).   *For any given $f \in \mathcal{F}_{CD}$ and $1 \le p \le \infty$, there exists a $\delta > 0$ such that there exists a piece-wise constant function $\bar{f}$ with $d_p(f, \bar{f}) \le \frac{\epsilon}{3}$. Concretely, $\bar{f}$ is defined as*

$$\bar{f}(X) = \sum_{P \in \mathbb{G}_\delta} f(P) \cdot \mathbf{1}\left[ \|\text{ReLU}(X - P)\|_\infty \le \delta \right]$$

Since transformers can learn a positional embedding $E$, without any loss of generality, we can consider the translated function. In particular, define

$$E = \begin{bmatrix} 0 & 0 & 0 & \dots & 0 \\ \delta^{-d} & \delta^{-d} & \delta^{-d} & \dots & \delta^{-d} \\ \delta^{-2d} & \delta^{-2d} & \delta^{-2d} & \dots & \delta^{-2d} \\ \vdots & & & & \\ \delta^{-(n-1)d} & \delta^{-(n-1)d} & \delta^{-(n-1)d} & \dots & \delta^{-(n-1)d} \end{bmatrix}$$

We will try to approximate $g(X) = f(X - E)$ where $g$ is defined on the domain $[0,1]^d \times [\delta^{-d}, \delta^{-d} + 1]^d \times \cdots \times [\delta^{-(n-1)d}, \delta^{-(n-1)d} + 1]^d$. To do so, we will apply a suitable modification of Lemma 1, which will consider the discretized grid

$$\mathbf{G}_\delta^E := [0,1]_\delta^d \times [\delta^{-d}, \delta^{-d} + 1]_\delta^d \times \cdots \times [\delta^{-(n-1)d}, \delta^{-(n-1)d} + 1]_\delta^d.$$

Therefore, it suffices to approximate a function $\bar{f} : \mathbf{G}_\delta^E \to \mathbb{R}^{n \times d}$ defined as

$$\bar{f}(X) = \sum_{P \in \mathbf{G}_\delta^E} f(P - E) \cdot \mathbf{1}\left[ \|\mathrm{ReLU}(X - P)\|_\infty \leq \delta \right].$$

### A.2.2 Contextual Mappings and Sparse Attention Mechanisms

Throughout this section, we will assume that we are given a function that has an extra global token at index $0$ and all vectors have an extra dimension appended to them. The latter assumption is without loss of generality as we can use the Feed-Forward Network to append sparse dimensions. In particular, we will associate $X \in \mathbb{R}^{(n+1) \times (d+1)}$ where we write $X = (x_0, x_1, \dots, x_n)$. Although our function is only defined for $\mathbf{G}_\delta^E \subset \mathbb{R}^{n \times d}$, we can amend the function in a natural way by making it ignore the first column. To avoid excessive clutter, we will assume that the function value is evaluated on the last $n$ columns.

The main idea in this section is the use of contextual mapping to enable Transformers to compute any discretized function. A contextual mapping is an unique encoding of each tuple $(X, x_i)$ where $X \in \mathbf{G}_\delta^E$, and each column $x_i \in [\delta^{-(i-1)d}, \delta^{-(i-1)d} + 1)_\delta^d$ for all $i \in [n]$. We restate the definition adapted to our setting below

**Definition 2** (Defn 3.1 [105]). *(Contextual Mapping) A contextual mapping is a function mapping $q : \mathbf{G}_\delta^E \to \mathbb{R}^n$ if it satisfies the following:*

1. *For any $P \in \mathbf{G}_\delta^E$, $q(P)$ contains distinct entries.*

2. *For any two $P, P' \in \mathbf{G}_\delta^E$ with $P \neq P'$, all entries of $q(P)$ and $q(P')$ are distinct.*

The key technical novelty of the proof is computing a contextual mapping using only the sparse attention mechanism. We create a "selective shift" operator which only shifts entries of a vector that lie in a certain range. We will use this shift operator strategically to ensure that we attain a contextual mapping at the end of the process. The lemma below, which is based on parts of the proof of Lemma 6 of [105], states that we can implement a suitable "selective" shift operator using a sparse attention mechanism.

**Lemma 2.** *Given a function $\psi : \mathbb{R}^{(n+1) \times (d+1)} \times \mathbb{R}^2 \to \mathbb{R}^{(n+1) \times 1}$ and a vector $u \in \mathbb{R}^{d+1}$ and a sparse attention mechanism based on the directed graph $D$, we can implement a selective shift operator that receives as input a matrix $X \in \mathbb{R}^{(n+1) \times (d+1)}$ and outputs $X + \rho \cdot \psi_u(X, b_1, b_2)$ where*

$$\psi_u(Z; b_1, b_2)_i = \begin{cases} (\max_{j \in N(i)} u^T Z_j - \min_{j \in N(i)} u^T Z_j)e_1 & \text{if } b_1 \leq u^T Z_j \leq b_2 \\ 0 & \text{else.} \end{cases}$$

*Note that $e_1 \in R^{d+1}$ denotes $(1, 0, \dots, 0)$.*

*Proof.* Consider the function , which can be implemented by a sparse attention mechanism :

$$\tilde{\psi}(X, b)_i = \sigma_H \left[ (u^T \cdot X_i)^T \cdot (u^T X_{N(i)} - b1_{N(i)}^T)e^{(1)}(u^T X_{N(i)}) \right]$$

This is because the Key, Query and Value functions are simply affine transformations of $X$.

Given any graph $D$, the above function will evaluate to the following:

$$\tilde{\psi}(Z;b)_i = \begin{cases} (\max_{j \in N(i)} u^T Z_j)e_1 & \text{if } u^T Z_j > b \\ (\min_{j \in N(i)} u^T Z_j)e_1 & \text{if } u^T Z_j < b \end{cases}$$

Therefore we can say that $\tilde{\psi}(Z;b_Q) - \tilde{\psi}(Z;b_{Q'})$ satisfies

$$\psi(Z;b_1,b_2)_i = \begin{cases} (\max_{j \in N(i)} u^T Z_j - \min_{j \in N(i)} u^T Z_j)e_1 & \text{if } b_1 \leq u^T Z_j \leq b_2 \\ 0 & \text{else} \end{cases}$$

$\square$

The following lemma, which is the heart of the proof, uses the above selective shift operators to construct contextual mappings.

**Lemma 3.** *There exists a function $g_c : \mathbb{R}^{(n+1) \times (d+1)} \to \mathbb{R}^{(n+1)}$ and a unique vector $u$, such that for all $P \in \mathbf{G}_\delta^E$ $g_c(P) := \langle u, g(P) \rangle$ satisfies the property that $g_c$ is a contextual mapping of $P$. Furthermore, $g_c \in \mathcal{T}_D^{2,1,1}$ using a composition of sparse attention layers as long as $D$ contains the star graph.*

*Proof.* Define $u \in \mathbb{R}^{d+1} = [1, \delta^{-1}, \delta^{-2}, \ldots, \delta^{-d+1}, \delta^{-nd}]$ and let $X_0 = (0, \ldots, 0, 1)$. We will assume that $\langle x_i, x_0 \rangle = 0$, by assuming that all the columns $x_1, \ldots, x_n$ are appended by 0.

To successfully encode the entire context in each token, we will interleave the shift operator to target the original columns $1, \ldots, n$ and to target the global column 0. After a column $i$ is targeted, its inner product with $u$ will encode the entire context of the first $i$ columns. Next, we will shift the global token to take this context into account. This can be subsequently used by the remaining columns.

For $i \in \{0, 1, \ldots, n\}$, we will use $l_i$ to denote the innerproducts $\langle u, x_i \rangle$ at the beginning. For $f_i = \langle u, x_i \rangle$ after the $i^{th}$ column has changed for $i \in \{1, \ldots, n\}$ and we will use $f_0^k$ to denote $\langle u, x_0 \rangle$ after the $k^{th}$ phase. We need to distinguish the global token further as it's inner product will change in each phase. Initially, given $X \in \mathbf{G}_\delta^E$, the following are true:

$$\delta^{-(i-1)d} \leq \langle u, X_i \rangle \leq \delta^{-id} - \delta \qquad \text{for all } i \in [n]$$

$$\delta^{-(n+1)d} = \langle u, X_0 \rangle$$

Note that all $l_i$ ordered in distinct buckets $l_1 < l_2 < \cdots < l_n < l_0$.

We do this in phases indexed from $i \in \{1, \ldots, n\}$. Each phase consists of two distinct parts:
**The low shift operation:** These operation will be of the form

$$X \leftarrow X + \delta^{-d}\psi\left(X, v - \delta/2, v + \delta/2\right)$$

for values $v \in [\delta^{-id}), \delta^{-(i+1)d})_\delta$. The range is chosen so that only $l_i$ will be in the range and no other $l_j$ $j \neq i$ is in the range. This will shift exactly the $i^{th}$ column $x_i$ so that the new inner product $f_i = \langle u, x_i \rangle$ is substantially larger than $l_i$. Furthermore, no other column of $X$ will be affected.
**The high shift operation:** These operation will be of the form

$$X \leftarrow X + \delta^{-nd} \cdot \psi\left(X, v - \delta/2, v + \delta/2\right)$$

for values $v \in [S_i, T_i)_\delta$. The range $[S_i, T_i)_\delta$ is chosen to only affect the column $x_0$ (corresponding to the global token) and no other column. In particular, this will shift the global token by a further $\delta^{-nd}$. Let $\tilde{f}_0^i$ denote the value of $\tilde{f}_0^i = \langle u, x_0 \rangle$ at the end of $i^{th}$ high operation.

Each phase interleaves a shift operation to column $i$ and updates the global token. After each phase, the updated $i^{th}$ column $f_i = \langle u, x_i \rangle$ will contain a unique token encoding the values of all the $l_1, \ldots, l_i$. After the high update, $\tilde{f}_0^i = \langle u, x_0 \rangle$ will contain information about the first $i$ tokens.

Finally, we define the following constants for all $k \in \{0, 1, \ldots, n\}$.

$$T_k = (\delta^{-(n+1)d} + 1)^k \cdot \delta^{-nd} - \sum_{t=2}^{k} (\delta^{-(n+1)d} + 1)^{k-t} (2\delta^{-nd-d} + \delta^{-nd} + 1)\delta^{-td}$$

$$- (\delta^{-(n+1)d} + 1)^{k-1}(\delta^{-nd-d} + \delta^{-nd})\delta^{-d} - \delta^{-(k+1)d} \tag{UP}$$

$$S_k = (\delta^{-(n+1)d} + 1)^k \cdot \delta^{-nd} - \sum_{t=2}^{k} (\delta^{-(n+1)d} + 1)^{k-t}(2\delta^{-nd-d} + \delta^{-nd} + 1)\delta^{-(t-1)d}$$
$$- (\delta^{-(n+1)d} + 1)^{k-1}(\delta^{-nd-d} + \delta^{-nd}) - \delta^{-kd} \tag{LP}$$

After each $k$ phases, we will maintain the following invariants:

1. $S_k < \tilde{f}_0^k < T_k$ for all $k \in \{0, 1, \dots, n\}$.

2. $T_{k-1} \le f_k < S_k$

3. The order of the inner products after $k^{th}$ phase is

$$l_{k+1} < l_{k+2} \cdots < l_n < f_1 < f_2 < \cdots < f_k < \tilde{f}_0^k.$$

**Base case** The case $k = 0$, is trivial as we simply set $S_0 = \delta^{-(n+1)d}$, $T_0 = \delta^{-(n+1)\cdot d} + \delta$.
The first nontrivial case is $k = 1$.

**Inductive Step** First, in the low shift operation is performed in the range $[\delta^{-(k-1)d}, \delta^{-kd})_\delta$ Due to the invariant, we know that there exists only one column $x_k$ that is affected by this shift. In particular, for column $k$, we will have $\max_{j \in N(k)} \langle u, x_j \rangle = \langle u, x_0 \rangle = \tilde{f}_0^{k-1}$. The minimum is $l_k$. Thus the update will be $f_k = \delta^{-d}(\tilde{f}_0^{k-1} - l_k) + l_k$. Observe that for small enough $\delta$, $f_k \ge \tilde{f}_0^{k-1}$. Hence the total ordering, after this operation is

$$l_k + 1 < l_{k+2} \cdots < l_n < f_1 < f_2 < \cdots < \tilde{f}_0^{k-1} < f_k \tag{2}$$

Now when we operate a higher selective shift operator in the range $[S_{k-1}, T_{k-1})_\delta$. Since only global token's innerproduct $\tilde{f}_0^{k-1}$ is in this range, it will be the only column affected by the shift operator. The global token operates over the entire range, we know from Eq. (2) that, $f_k = \max_{i \in [n]} \langle u, x_i \rangle$ and $l_{k+1} = \min_{i \in [n]} \langle u, x_i \rangle$. The new value $\tilde{f}_0^k = \delta^{-nd} \cdot (f_k - l_{k+1}) + \tilde{f}_0^{k-1}$. Expanding and simplifying we get,

$$\begin{aligned}
\tilde{f}_0^k &= \delta^{-nd} \cdot (f_k - l_{k+1}) + \tilde{f}_0^{k-1} \\
&= \delta^{-nd} \cdot (\delta^{-d}(\tilde{f}_0^{k-1} - l_k) + l_k - l_{k+1}) + \tilde{f}_0^{k-1} \\
&= \delta^{-(n+1)d} \cdot (\tilde{f}_0^{k-1} - l_k) + \delta^{-nd}(l_k - l_{k+1}) + \tilde{f}_0^{k-1} \\
&= (\delta^{-(n+1)d} + 1)\tilde{f}_0^{k-1} - (\delta^{-nd-d} + \delta^{-nd})l_k - l_{k+1}
\end{aligned}$$

Expanding the above recursively, we get

$$= (\delta^{-(n+1)d} + 1)^k \cdot \tilde{f}_0^0 - \sum_{t=2}^{k} (\delta^{-(n+1)d} + 1)^{k-t}(2\delta^{-nd-d} + \delta^{-nd} + 1)l_t$$
$$- (\delta^{-(n+1)d} + 1)^{k-1}(\delta^{-nd-d} + \delta^{-nd})l_1 - l_{k+1}$$

Since we know that $\tilde{f}_0^0 = \delta^{-nd}$ and each $l_i < \delta^{-id}$, we can substitute this to get Eq. (UP) and we can get an lower-bound Eq. (LP) by using $l_i \ge \delta^{-(i-1)d}$.

By construction, we know that $S_k \le \tilde{f}_0^k < T_k$. For sufficiently small $\delta$, observe that $S_k \le \tilde{f}_0^k < T_k$ all are essentially the dominant term $\approx O(\delta^{-n(k+1)d-kd})$ and all the lower order terms do not matter. As a result it is immediate to see that that $f_k > \delta^{-d}(\tilde{f}_0^{k-1} - l_k) > T_{k-1}$ and hence we can see that the invariant 2 is also satisfied. Since only column $k$ and the global token are affected, we can see that invariant 3 is also satisfied.

After $n$ iterations, $\tilde{f}_0^n$ contains a unique encoding for any $P \in \mathbf{G}_\delta^E$. To ensure that all tokens are distinct, we will add an additional layer $X = X + \delta^{-n^2 d}\psi(X, v - \delta/2, v + \delta/2)$ for all $v \in [S_1, T_n)_\delta$. This ensures that for all $P, P' \in \mathbf{G}_\delta^E$, each entry of $q(P)$ and $q(P')$ are distinct. $\square$

The previous lemma shows that we can compute a contextual mapping using only sparse transforms. We now use the following lemma to show that we can use a contextual mapping and feed-forward layers to accurately map to the desired output of the function $\bar{f}$.

**Lemma 4** (Lemma 7 [105]). *Let $g_c$ be the function in Lemma 3, we can construct a function $g_v : \mathbb{R}^{(n+1)\times(d+1)} \to \mathbb{R}^{(n+1)\times d}$ composed of $O(n\delta^{-nd})$ feed-forward layers (with hidden dimension $q = 1$) with activations in $\Phi$ such that $g_v$ is defined as $g_v(Z) = [g_v^{tkn}(Z_1), \ldots, g_v^{tkn}(Z_n)]$, where for all $j \in \{1, \ldots, n\}$,*

$$g_v^{tkn}(g_c(L)_j) = f(L)_j$$

### A.2.3 Approximating modified Transformers by Transformers

The previous section assumed we used Transformers that used hardmax operator $\sigma_H$ and activations functions belonging to the set $\Phi$. This is without loss of generality as following lemma shows.

**Lemma 5** (Lemma 9 [105]). *For each $g \in \bar{\mathcal{T}}^{2,1,1}$ and $1 \leq p \leq \infty$, $\exists \bar{g} \in \mathcal{T}^{2,1,4}$ such that $d_p(g, \bar{g}) \leq \epsilon/3$*

Combining the above lemma with the Lemma 3, we get our main result:

**Theorem 2.** *Let $1 \leq p \leq \infty$ and $\epsilon > 0$, there exists a transformer network $g \in \mathcal{T}_D^{2,1,4}$ which achieves a ratio of $d_p(\bar{f}, g) \leq \epsilon$ where $D$ is the sparse graph.*

Since the sparsity graph associated with BIGBIRD contains a star network, we know that it can express any continuous function from a compact domain.

**Contemporary work on Universal Approximability of Sparse Transformers** We would like to note that, contemporary work done by Yun et al. [106], also parallelly explored the ability of sparse transformers with linear connections to capture sequence-to-sequence functions on the compact domain.

# B  Turing Completeness

In this section, we will extend our results to the setting of Pérez et al. [72]. Our exposition will largely use their proof structure but we will make a few changes. We repeat some of the lemmas with the amendments to make the exposition self-contained.

## B.1  Notation

**Transformer Decoder**   We need both an encoder and a decoder in the transformer for simulating a Turing machine. We utilize the same notation used in App. A.1 for encoders. The decoder is similar to an encoder but with additional attention to an external pair of key-value vectors ($\boldsymbol{K}^{\mathbf{e}} \in \mathbb{R}^{n \times m}, \boldsymbol{V}^{\mathbf{e}} \in \mathbb{R}^{n \times d}$), which usually come from the encoder stack. A single layer of Transformer decoder is a parametric function Dec receiving a sequence $\boldsymbol{Y}_j = (\boldsymbol{y}_1, \ldots, \boldsymbol{y}_j)$ of vectors in $\mathbb{R}^d$ plus the external ($\boldsymbol{K}^{\mathbf{e}}, \boldsymbol{V}^{\mathbf{e}}$) and returning a sequence of vectors $\boldsymbol{Z}_j = (\boldsymbol{z}_1, \ldots, \boldsymbol{z}_j)$ of the same length. Each $\boldsymbol{z}_i$ is a $d$ dimensional vector as well. Dec has three components, one more than Enc:

1. An attention mechanism ATTN that takes in the sequence $\boldsymbol{Y}_j$ and returns sequence $(\boldsymbol{p}_1, ..., \boldsymbol{p}_j)$ of the same length and dimensionality;

2. A cross-attention mechanism CROSSATTN that takes in the sequence $(\boldsymbol{p}_1, ..., \boldsymbol{p}_j)$ plus the external ($\boldsymbol{K}^{\mathbf{e}}, \boldsymbol{V}^{\mathbf{e}}$) and returns sequence $(\boldsymbol{a}_1, ..., \boldsymbol{a}_j)$, with each $\boldsymbol{a}_i \in \mathbb{R}^d$; and

3. A two layer fully connected network $O$ that takes in a vector in $\mathbb{R}^d$ and returns a vector in $\mathbb{R}^d$.

Then $i$-th output vector of $\text{Dec}(\boldsymbol{Y}_j; \boldsymbol{K}^{\mathbf{e}}, \boldsymbol{V}^{\mathbf{e}})$ is computed as follows:

$$\boldsymbol{z}_i = O(\boldsymbol{a}_i) + \boldsymbol{a}_i \tag{3}$$

where
$$\boldsymbol{a}_i = \text{CROSSATTN}(\boldsymbol{p}_i, \boldsymbol{K}^{\mathbf{e}}, \boldsymbol{V}^{\mathbf{e}}) + \boldsymbol{p}_i \tag{4}$$

and
$$\boldsymbol{p}_i = \text{ATTN}_D(\boldsymbol{Y}_j)_i + \boldsymbol{y}_i \tag{5}$$

$\text{ATTN}_D$ and $O$ are as defined in App. A.1 and it remains to define CROSSATTN. The $i^{\text{th}}$ output vector of multi-head cross-attention attention is given by

$$\text{CROSSATTN}(\boldsymbol{Y}_j)_i = \sum_{h=1}^{H} \sigma\left((\boldsymbol{y}_i W_Q^h)(\boldsymbol{K}^{(e)} W_K^h)^T\right) \cdot (\boldsymbol{V}^{(e)} W_V^h) \tag{6}$$

where $W_Q^h, W_K^h, W_V^h \in \mathbb{R}^{d \times m}, W_V^h \in \mathbb{R}^{d \times d}$, for all $h = 1, \ldots H$ heads.

**Turning Machine**   We will use the same setup of Turning Machine that was used by Pérez et al. [72] (see section B.4). Given a Turing Machine $M = (Q, \Sigma, \delta, q_{init}, F)$, we use the following notation

$$q^{(j)} : \text{ state of Turing machine } M \text{ at time } j.$$
$$s^{(j)} : \text{ symbol under the head of } M \text{ at time } j.$$
$$v^{(j)} : \text{ symbol written by } M \text{ at time } j.$$
$$m^{(j)} : \text{ head direction in the transition of } M \text{ at time } j.$$

**Vector representations**   For a symbol $s \in \Sigma$, $[\![ s ]\!]$ denotes its one-hot vector representation in $\mathbb{Q}^{|\Sigma|}$. All the transformer intermediate vectors used in our simulations have dimension $d = 2|Q| + 4|\Sigma| + 16$. Note that we use five extra dimension as compared to Pérez et al. [72]. We follow the convention used in Pérez et al. [72] and write a a vector $\boldsymbol{v} \in \mathbb{Q}^d$ arranged in four groups of values as follows

$$\boldsymbol{v} = [\quad \boldsymbol{q}_1, \boldsymbol{s}_1, x_1, \\ \boldsymbol{q}_2, \boldsymbol{s}_2, x_2, x_3, x_4, x_5, x_6, \\ \boldsymbol{s}_3, x_7, \boldsymbol{s}_4, \\ x_8, x_9, x_{10}, x_{11}, x_{12}, x_{13}, x_{14}, x_{15}, x_{16} \quad ]$$

where $\boldsymbol{q}_i \in \mathbb{Q}^{|Q|}$, $\boldsymbol{s}_i \in \mathbb{Q}^{|\Sigma|}$, and $x_i \in \mathbb{Q}$.

## B.2  Details of the Simulation

In this section, we give more details on the architecture of the encoder and decoder needed to implement our simulation strategy.

**High Level Overview:** Given the Turing machine $M$, we will show that a transformer with an appropriate encoder and decoder $\mathcal{T}_D$ can simulate each step of $M$'s execution. Our simulation strategy will mostly follow Pérez et al. [72], except we will use a sparse attention mechanism. The main idea is to maintain the current Turing machine state $q^{(j)}$ and symbol under the head $s^{(j)}$ as part of the decoder sequence $\boldsymbol{Y}$ for all time step $j$ so that we can always simulate the corresponding Turing machine transition $\delta(q^{(j)}, s^{(j)}) = (q^{(j)}, v^{(j)}, m^{(j)})$. The key difference will rise in Lemma B.4 of Pérez et al. [72], where full attention is used to select the appropriate symbol from tape history in one step. To accomplish the same task with sparse attention, we will exploit the associative property of max and break down the symbol selection over multiple steps. Thus, unlike Pérez et al. [72] one decoding step of our sparse transformer $\mathcal{T}_D$ does not correspond to one step of the Turing machine $M$. In particular, we will have two type of steps: compute step corresponding to update of $M$'s state and intermediate steps corresponding to aggregating the max (which in turn is used for symbol selection). Let $i$ denote the step of $\mathcal{T}_D$ and $g(i)$ denote the step of $M$ being simulated at step $i$ of the decoder. At each decoding step we want to maintain the current Turing machine state $q^{g(i)}$ and symbol under the $s^{g(i)}$ in $\boldsymbol{y}_i$. For roughly $O(\sqrt{i})$ intermediate steps the state will remain the same, while we aggregate information about relevant past output symbols through sparse attention. To maintain the same state for intermediate steps, we introduce an extra switching layer (App. B.2.3). Finally, at the next compute step we will make the transition to new state $q^{g(i)+1}$, new head movement $m^{g(i)}$, and new output symbol $v^{g(i)}$ to be written. Thereby we are able to completely simulate the given Turing machine $M$. As a result, we can prove the following main theorem:

**Theorem 3.** *There exists a sparse attention mechanism using $O(n)$ inner products such that the resulting class of Transformer Networks using this sparse attention mechanism is Turing Complete.*

### Encoder

As [72], we use the same trivial single layer encoder where resulting $\boldsymbol{K}^{(e)}$ contains position embedding and $\boldsymbol{V}^{(e)}$ contains one-hot symbol representation.

### Decoder

**Sparse Self-Attention mechanism for Decoder** In this section, we will consider a particular instance of the sparse graph $D$ at decoder. We define its edges to be given by the following relations: $\forall j \in \mathbb{N}_+, 1 \leq k \leq j+1$,

$$\left(\frac{j(j+1)}{2} + k, \frac{k(k+1)}{2}\right) \text{ and }$$

$$\left(\frac{j(j+1)}{2} + k, \frac{j(j+1)}{2} + k\right) \text{ if } k > 1 \text{ else } \left(\frac{j(j+1)}{2} + 1, \frac{j(j+1)}{2}\right).$$

This graph can be seen as a special case of BIGBIRD where first type of edges are realizations of random and second type of edges correspond to locality. Also note that this graph satisfies the left-to-right constraint of decoder, i.e. no node attends to a node in the future.

| Transform i: | 1 | 2 | 3 | 4 | 5 | 6 | 7 | 8 | 9 | 10 | 11 | 12 | 13 | 14 | 15 |
|---|---|---|---|---|---|---|---|---|---|---|---|---|---|---|---|
| TM Step j: | 0 | 1 | 1 | 2 | 2 | 2 | 3 | 3 | 3 | 3 | 4 | 4 | 4 | 4 | 4 |
| Offset k: | 1 | 1 | 2 | 1 | 2 | 3 | 1 | 2 | 3 | 4 | 1 | 2 | 3 | 4 | 5 |

Figure 2: Mapping between transformer step and original Turing machine step.

**Embeddings and positional encodings**  Our construction needs a different positional encoding $\text{pos}_{\text{Dec}} : \mathbb{N} \to \mathbb{Q}^d$ for decoder:

$$
\text{pos}_{\text{Dec}}(i) \;=\; [ \quad 0,\ldots,0, \\
0,\ldots,0, \\
0,\ldots,0, \\
1, g(i)+1, \tfrac{1}{g(i)+1}, \tfrac{1}{(g(i)+1)^2}, h(i), 0, 0, 0, 0 \quad ]
$$

where $g(i) = \left\lfloor \frac{-1+\sqrt{1+8i}}{2} \right\rfloor$ and $h(i) = g(i+1) - g(i)$. Note that $h(i)$ reduces to a binary indicator variable $\mathbf{1}\left\{ \frac{-1+\sqrt{1+8i}}{2} = \left\lfloor \frac{-1+\sqrt{1+8i}}{2} \right\rfloor \right\}$.

**Induction Setup**

We next show how to construct the decoder layers to produce the sequence of outputs $\boldsymbol{y}_1, \boldsymbol{y}_2, \ldots$, where $\boldsymbol{y}_i$ is given by:

$$
\boldsymbol{y}_i \;=\; [ \quad [\![\, q^{g(i)} \,]\!], [\![\, s^{g(i)} \,]\!], c^{g(i)}, \\
0,\ldots,0, \\
\mathbf{0}_s, 0, [\![\, w^{(i)} \,]\!], \\
0,0,0,0,0, u_1^{(i)}, u_2^{(i)}, u_3^{(i)}, u_4^{(i)} \quad ]
$$

That is, at step $i$ of our sparse decoder $\boldsymbol{y}_i$, it will contain the information about the state of the turing machine $M$ at time $g(i)$, the symbol under the head of $M$ at time $g(i)$, and the current location of head of $M$ at time $g(i)$. We also have a placeholder symbol $w$ and placeholder scalars $u_1, u_2, u_3$, whose role will be clear from our construction.

We consider as the starting vector for the decoder the vector

$$
\boldsymbol{y}_1 \;=\; [ \quad [\![\, q_{\text{init}} \,]\!], [\![\, \# \,]\!], 0, \\
0,\ldots,0, \\
0,\ldots,0, \\
0,\ldots,0 \quad ]
$$

We assume that the start head is at $c^{(0)} = 0$, the initial state is $q^{(0)} = q_{\text{init}}$, and $s^{(0)} = \#$ as we initialize from clean tape. We show the correctness of our construction by an inductive argument: we describe the architecture piece by piece and at the same time will show for every $r \geq 0$, our architecture constructs $\boldsymbol{y}_{r+1}$ from the previous vectors $(\boldsymbol{y}_0, \ldots, \boldsymbol{y}_r)$.

Thus, assume that $\boldsymbol{y}_1, \ldots, \boldsymbol{y}_r$ satisfy the properties stated above. Since we are using positional encodings, the actual input for the first layer of the decoder is the sequence

$$
\boldsymbol{y}_1 + \text{pos}_{\text{Dec}}(1), \; \boldsymbol{y}_2 + \text{pos}_{\text{Dec}}(2), \; \ldots, \; \boldsymbol{y}_r + \text{pos}_{\text{Dec}}(r).
$$

We denote by $\overline{\boldsymbol{y}}_i$ the vector $\boldsymbol{y}_i$ plus its positional encoding. Thus we have $\forall \, 1 \leq i \leq r$ that

$$
\overline{\boldsymbol{y}}_i \;=\; [ \quad [\![\, q^{g(i)} \,]\!], [\![\, s^{g(i)} \,]\!], c^{g(i)}, \\
0,\ldots,0, \\
\mathbf{0}_s, 0, [\![\, w^{(i)} \,]\!], \\
1, g(i)+1, \tfrac{1}{g(i)+1}, \tfrac{1}{(g(i)+1)^2}, h(i), u_1^{(i)}, u_2^{(i)}, u_3^{(i)}, u_4^{(i)} \quad ]
$$

### B.2.1   Layer 1: Simulate Transition Function

In this layer, we use the cross-attention between encoder and decoder to access the input string and a feed-forward network to simulate the transition function of $M$. The first self attention in Eq. (5) is not used in this layer and we just produce the identity. This identity function is achieved by setting all queries, keys, values to be 0 everywhere plus the residual connection. Thus, we have $\boldsymbol{p}_i^1 = \overline{\boldsymbol{y}}_i$.

Since $\boldsymbol{p}_i^1$ is of the form $[\_,\ldots,\_, 1, g(i)+1, \_, \ldots, \_]$, we know by Lemma B.1 of Pérez et al. [72] that if we use $\boldsymbol{p}_i^1$ to attend over the encoder we obtain

$$
\text{CROSSATTN}(\boldsymbol{p}_i^1, \boldsymbol{K}^{\mathbf{e}}, \boldsymbol{V}^{\mathbf{e}}) \;=\; [ \quad 0,\ldots,0, \\
0,\ldots,0, \\
[\![\, \alpha^{g(i)+1} \,]\!], \beta^{g(i)+1}, \mathbf{0}_s, \\
0,\ldots,0 \quad ]
$$

where $\alpha$ and $\beta$ are as defined in Eq. (21) of [72]. Thus in Eq. (4) we finally produce the vector $\boldsymbol{a}_i^1$ given by

$$
\begin{aligned}
\boldsymbol{a}_i^1 &= \text{CROSSATTN}(\boldsymbol{p}_i^1, \boldsymbol{K}^{\mathbf{e}}, \boldsymbol{V}^{\mathbf{e}}) + \boldsymbol{p}_i^1 \\
&= [ \quad [\![\, q^{g(i)} \,]\!], [\![\, s^{g(i)} \,]\!], c^{g(i)}, \\
&\qquad 0, \ldots, 0, \\
&\qquad [\![\, \alpha^{g(i)+1} \,]\!], \beta^{g(i)+1}, [\![\, w^{(i)} \,]\!], \\
&\qquad 1, g(i)+1, \tfrac{1}{g(i)+1}, \tfrac{1}{(g(i)+1)^2}, h(i), u_1^{(i)}, u_2^{(i)}, u_3^{(i)}, u_4^{(i)} \quad ]
\end{aligned}
\tag{7}
$$

As the final piece of the first decoder layer we use a function $O_1(\cdot)$ (Eq. (3)) that satisfies the following lemma.

**Lemma 6** (Lemma B.2 [72]). *There exists a two-layer feed-forward network $O_1 : \mathbb{Q}^d \to \mathbb{Q}^d$ such that with input vector $\boldsymbol{a}_i^1$ (Eq. (7)) produces as output*

$$
\begin{aligned}
O_1(\boldsymbol{a}_i^1) &= [ \quad 0, \ldots, 0, \\
&\qquad [\![\, q^{g(i)+1} \,]\!], [\![\, v^{g(i)} \,]\!], m^{g(i)}, 0, 0, 0, 0 \\
&\qquad 0, \ldots, 0, \\
&\qquad 0, \ldots, 0 \qquad\qquad\qquad\qquad\qquad ]
\end{aligned}
$$

*That is, function $O_1(\cdot)$ simulates transition $\delta(q^{g(i)}, s^{g(i)})$ to construct $[\![\, q^{g(i)+1} \,]\!]$, $[\![\, v^{g(i)} \,]\!]$, and $m^{g(i)}$ besides some other linear transformations.*

Thus, finally the output of the first decoder layer is

$$
\begin{aligned}
\boldsymbol{z}_i^1 = O_1(\boldsymbol{a}_i^1) + \boldsymbol{a}_i^1 &= [ \quad [\![\, q^{g(i)} \,]\!], [\![\, s^{g(i)} \,]\!], c^{g(i)}, \\
&\qquad [\![\, q^{g(i)+1} \,]\!], [\![\, v^{g(i)} \,]\!], m^{g(i)}, 0, 0, 0, 0, \\
&\qquad [\![\, \alpha^{g(i)+1} \,]\!], \beta^{g(i)+1}, [\![\, w^{(i)} \,]\!], \\
&\qquad 1, g(i)+1, \tfrac{1}{g(i)+1}, \tfrac{1}{(g(i)+1)^2}, h(i), u_1^{(i)}, u_2^{(i)}, u_3^{(i)}, u_4^{(i)} \quad ]
\end{aligned}
$$

### B.2.2 Layer 2: Finding Head Node

In this layer, we only use the feed-forward network to evaluate the next location of the head. The self-attention and cross-attention are set to be the identity function, so $\boldsymbol{a}_i^2 = \boldsymbol{p}_i^2 = \boldsymbol{z}_i^1$. Recall that $c^{g(i)}$ is the cell to which $M$ is pointing to at time $g(i)$, and that it satisfies the following recursion $c^{g(i)+1} = c^{g(i)} + m^{g(i)}$, which can be expanded to see that that $c^{g(i)+1} = m^{(0)} + m^{(1)} + \cdots + m^{g(i)}$. Its not difficult to see that a two layer network with non-linearity can compute $c^{g(i)+1}/(g(i)+1)$ and $c^{g(i)}/(g(i)+1)$ from $c^{g(i)}$, $m^{g(i)}$, and $1/(g(i)+1)$ using the relation $c^{g(i)+1} = c^{g(i)} + m^{g(i)}$. At the end of layer 2, we obtain

$$
\begin{aligned}
\boldsymbol{z}_i^2 = O_2(\boldsymbol{a}_i^2) + \boldsymbol{a}_i^2 &= [ \quad [\![\, q^{g(i)} \,]\!], [\![\, s^{g(i)} \,]\!], c^{g(i)}, \\
&\qquad [\![\, q^{g(i)+1} \,]\!], [\![\, v^{g(i)} \,]\!], c^{g(i)+1}, \tfrac{1}{g(i)+1}, \tfrac{1}{(g(i)+1)^2}, \tfrac{c^{g(i)+1}}{g(i)+1}, \tfrac{c^{g(i)}}{g(i)+1}, \\
&\qquad [\![\, \alpha^{g(i)+1} \,]\!], \beta^{g(i)+1}, [\![\, w^{(i)} \,]\!], \\
&\qquad 1, g(i)+1, \tfrac{1}{g(i)+1}, \tfrac{1}{(g(i)+1)^2}, h(i), u_1^{(i)}, u_2^{(i)}, u_3^{(i)}, u_4^{(i)} \quad ]
\end{aligned}
$$

### B.2.3 Layer 3: Distinguishing Node Type

This is an additional layer (not present in the work of [72]), where we propagate computations in our sparse graph. In particular, we will use this layer to "compute" or accumulate state in intermediate nodes. We make this clear below. The self-attention and cross-attention are all set to be the identity function, so $\boldsymbol{a}_i^3 = \boldsymbol{p}_i^3 = \boldsymbol{z}_i^2$. In this layer, we only use the dense attention layers to select the newly computed states or to continue with previous states. Using idea similar to Lemma B.6 of [72], we can construct a dense network such that

$$
O([\boldsymbol{x}, \boldsymbol{y}, \boldsymbol{z}, b])) = \begin{cases} [\mathbf{0}, \mathbf{0}, \mathbf{0}, 0] & \text{if } b = 1, \\ [\mathbf{0}, \boldsymbol{z} - \boldsymbol{y}, -\boldsymbol{z}, 0] & \text{if } b = 0. \end{cases}
$$

The negatives are generated to offset results from skip connection. We utilize such network to switch Turing machine state and position embedding for intermediate steps to the values received from

previous time step and do nothing for compute nodes. We use $h(i)$ as the flipping bit $b$. Thus, at end of layer 3, we obtain

$$
\begin{aligned}
\boldsymbol{z}_i^3 \;=\; O_3(\boldsymbol{a}_i^3) + \boldsymbol{a}_i^3 \;=\; [ \quad & 0,\dots,0, \\
& [\![\, \hat{q}^{(i)} \,]\!], [\![\, \hat{v}^{(i)} \,]\!], \hat{c}^{(i)}, \tfrac{1}{g(i)+1}, \tfrac{1}{(g(i)+1)^2}, \tfrac{c^{g(i)+1}}{g(i)+1}, \hat{u}_4^{(i)}, \\
& [\![\, \hat{\alpha}^{(i)} \,]\!], \hat{\beta}^{(i)}, \boldsymbol{0}_s, \\
& 1, \hat{u}_1^{(i)}, \hat{u}_2^{(i)}, \hat{u}_3^{(i)}, h(i), 0, 0, 0, 0 \qquad\qquad\qquad ]
\end{aligned}
$$

where we used $h(i)$ for selecting old states. In particular,

- We copy the input state and head position as is for intermediate nodes. We do not need to transition to next Turing machine states in these nodes.

$$
\hat{q}^{(i)} = \begin{cases} q^{g(i)+1} & \text{if } h(i) = 1 \\ q^{g(i)} & \text{if } h(i) = 0 \end{cases}, \quad
\hat{v}^{(i)} = \begin{cases} v^{g(i)} & \text{if } h(i) = 1 \\ w^{(i)} & \text{if } h(i) = 0 \end{cases}, \quad
\hat{c}^{(i)} = \begin{cases} c^{g(i)+1} & \text{if } h(i) = 1 \\ c^{g(i)} & \text{if } h(i) = 0 \end{cases}.
$$

- To preserve the symbol under the head for intermediate nodes, we copy the previous symbol to $\alpha$ location and set $\beta = g(i) + 1$, as the symbol at $\alpha$ location will be copied as the symbol under head for next transformer step by the final transformation layer if $\beta = g(i) + 1$. Thus, we correctly preserve the previous symbol under head as Turing machine does not transition these nodes. For compute nodes, things happen as usual.

$$
\hat{\alpha}^{(i)} = \begin{cases} \alpha^{g(i)+1} & \text{if } h(i) = 1 \\ s^{g(i)} & \text{if } h(i) = 0 \end{cases}, \qquad
\hat{\beta}^{(i)} = \begin{cases} \beta^{g(i)+1} & \text{if } h(i) = 1 \\ g(i) + 1 & \text{if } h(i) = 0 \end{cases}.
$$

- Finally for the intermediate nodes, we copy the position embedding corresponding to current best symbol $w$, which is stored in $u_1, u_2, u_3$. For compute node, we let the position embedding correspond to current Turing machine step.

$$
\hat{u}_1^{(i)} = \begin{cases} g(i) + 1 & \text{if } h(i) = 1 \\ u_1^{(i)} & \text{if } h(i) = 0 \end{cases}, \qquad
\hat{u}_2^{(i)} = \begin{cases} \tfrac{1}{(g(i)+1)} & \text{if } h(i) = 1 \\ u_2^{(i)} & \text{if } h(i) = 0 \end{cases},
$$

$$
\hat{u}_3^{(i)} = \begin{cases} \tfrac{1}{(g(i)+1)^2} & \text{if } h(i) = 1 \\ u_3^{(i)} & \text{if } h(i) = 0 \end{cases}, \qquad
\hat{u}_4^{(i)} = \begin{cases} \tfrac{c^{g(i)}}{g(i)+1} & \text{if } h(i) = 1 \\ u_4^{(i)} & \text{if } h(i) = 0 \end{cases}.
$$

For further simplification note that $g(i+1) = g(i)$ if $h(i) = 0$ else $g(i) + 1$ when $h(i) = 1$. With this fact, we can conclude that $\hat{q}^{(i)} = q^{g(i+1)}$ and $\hat{c}^{(i)} = c^{g(i+1)}$. Thus, we can write,

$$
\begin{aligned}
\boldsymbol{z}_i^3 \;=\; [ \quad & 0,\dots,0, \\
& [\![\, q^{g(i+1)} \,]\!], [\![\, \hat{v}^{(i)} \,]\!], c^{g(i+1)}, \tfrac{1}{g(i)+1}, \tfrac{1}{(g(i)+1)^2}, \tfrac{c^{g(i)+1}}{g(i)+1}, \hat{u}_4^{(i)}, \\
& [\![\, \hat{\alpha}^{(i)} \,]\!], \hat{\beta}^{(i)}, \boldsymbol{0}_s, \\
& 1, \hat{u}_1^{(i)}, \hat{u}_2^{(i)}, \hat{u}_3^{(i)}, h(i), 0, 0, 0, 0 \qquad\qquad\qquad ]
\end{aligned}
$$

### B.2.4 Layer 4: Finding next symbol on tape

To find the symbol on tape under next head position $c^{g(i)+1}$, we try to find what was written last at the location $c^{g(i)+1}$. To facilitate this, following [72], we define $\ell(j)$ to be the last time (previous to $j$) in which $M$ was pointing to position $c^{(j)}$, or it is $j - 1$ if this is the first time that $M$ is pointing to $c^{(j)}$. Recall $j$ is the Turing machine step counter, which is different from sparse transformer step $i$. [72] could utilize full attention mechanism to find $v^{\ell(j+1)}$ at one go, but we have to do it over multiple steps owing to our sparse attention mechanism.

We use similar query, key, value functions as used for full attention by [72] $\forall i$:

$$
\begin{aligned}
Q_4(\boldsymbol{z}_i^3) \;=\; [ \quad & 0,\dots,0 \\
& 0,\dots,0, \\
& 0,\dots,0, \\
& 0, \tfrac{c^{g(i)+1}}{g(i)+1}, \tfrac{1}{g(i)+1}, \tfrac{1}{3(g(i)+1)^2}, 0, 0, 0, 0, 0 \quad ]
\end{aligned}
$$

$$K_4(\boldsymbol{z}_i^3) \;=\; [\;\; 0,\ldots,0$$
$$0,\ldots,0,$$
$$0,\ldots,0,$$
$$0,\hat{u}_2^{(i)},\hat{u}_4^{(i)},\hat{u}_3^{(i)},0,0,0,0,0 \qquad]$$
$$V_4(\boldsymbol{z}_i^3) \;=\; [\;\; 0,\ldots,0,$$
$$0,\ldots,0,$$
$$\boldsymbol{0}_s,0,[\![\,\hat{v}^{(i)}\,]\!],$$
$$0,0,0,0,0,\hat{u}_1^{(i)},\hat{u}_2^{(i)},\hat{u}_3^{(i)},\hat{u}_4^{(i)} \qquad]$$

It is clear that the three functions are linear transformations and thus they can be defined by feed-forward networks. Notice that the query vector is always formed using current time step position embedding, whereas key and value vectors are formed using copied over entries for intermediate nodes and using current entries only for compute node.

Pérez et al. [72] find the desired $v^{l(j+1)}$ as $v^{m(j)}$ using full attention, where

$$m(t) = \underset{m\in\{0,\ldots,t\}}{\arg\min}\; \chi_t^j = \underset{m\in\{0,\ldots,t\}}{\arg\min}\; |\langle Q_4(\boldsymbol{z}_j^3), K_4(\boldsymbol{z}_m^3)\rangle|$$

Note the minimization is only over Turing machine steps, i.e. over compute nodes in our case. We show below that we can estimates $m(j)$ by parts using sparse attention mechanism. The main idea is just to notice that minimization problem $\min_{m\in\{0,\ldots,t\}} \chi_t^j$ can be expressed as $\min\{\cdots\min\{\min\{\chi_0^j,\chi_1^j\},\chi_2^j\},...,\chi_t^j\}$ by the associativity property.

By definition of our graph $D$, at every intermediate node $i$ of the form $j(j+1)/2+k$, i.e. where $k>0$, $g(i)=j$ and $h(i)=0$, we will attend over node $k(k+1)/2$ and best till now copied from $i-1$. The node $k(k+1)/2$ is never an intermediate node as $h(k(k+1)/2)=1$ for all $k$ and in fact corresponds to Turing machine's step $k$. This will help us select the key and value corresponding to min between node $k(k+1)/2$ and $i-1$. In other words, at node $i$ of the form $j(j+1)/2+k$ we would have evaluated $m(k)$ and corresponding value selected:

$$w^{(j(j+1)/2+k+1)} = \hat{v}^{m(k-1)}$$

and similarly for $u$'s. So after going through all the intermediate nodes, finally at the next compute node, i.e. when $k=j+1$, we will obtain the minimum value over all of $0,1,...,j$. This implies at a compute node will be able to recover $\ell(g(i)+1)$ and its corresponding value as shown in Lemma B.4 of [72]. Then we have that $\boldsymbol{p}_i^4$ is given by

$$\boldsymbol{p}_i^4 \;=\; \text{ATTN}_D(\boldsymbol{Z}_i^3) + \boldsymbol{z}_i^3$$
$$=\; [\;\; 0,\ldots,0,$$
$$[\![\,q^{g(i+1)}\,]\!],[\![\,\hat{v}^{(i)}\,]\!],c^{g(i+1)},0,\tfrac{c^{g(i)+1}}{g(i)+1},\hat{u}_4^{(i)},$$
$$[\![\,\hat{\alpha}^{(i)}\,]\!],\hat{\beta}^{(i)},[\![\,w^{(i+1)}\,]\!],$$
$$1,\hat{u}_1^{(i)},\hat{u}_2^{(i)},\hat{u}_3^{(i)},h(i),u_1^{(i+1)},u_2^{(i+1)},u_3^{(i+1)},u_4^{(i+1)} \qquad] \qquad (8)$$

The cross-attention and feed-forward network are set to be identity, so $\boldsymbol{z}_i^4 = \boldsymbol{a}_i^4 = \boldsymbol{p}_i^4$.

### B.2.5 Final transformation

We finish our construction by using the final transformation function $F(\cdot)$ from the corresponding lemma from Pérez et al. [72], with a slight modification.

**Lemma 7** (Lemma B.5 [72]). *There exists a function $F : \mathbb{Q}^d \to \mathbb{Q}^d$ defined by a feed-forward network such that*

$$F(\boldsymbol{z}_r^4) \;=\; [\;\; [\![\,q^{g(r+1)}\,]\!],[\![\,s^{g(r+1))}\,]\!],c^{g(r+1)},$$
$$0,\ldots,0,$$
$$\boldsymbol{0}_s,0,[\![\,w^{(r+1)}\,]\!],$$
$$0,0,0,0,0,u_1^{(r+1)},u_2^{(r+1)},u_3^{(r+1)},u_4^{(r+1)}]$$
$$=\; \boldsymbol{y}_{r+1}$$

The modification is to let $w, u_1, u_2, u_3$ to pass through. This yields the desired input to transformer at next time step for both intermediate and compute node, thereby concluding our induction.

# C Limitations

Finally, we show that sparse attention mechanisms can not universally replace dense attention mechanisms, i.e. there is no free lunch. We demonstrate a natural task which can be solved by the full attention mechanism in $O(1)$-layers. However, under standard complexity theoretic assumptions, we show that this problem will require $\tilde{\Omega}(n)$-layers for any sparse attention layers with $\tilde{O}(n)$ edges (not just BIGBIRD). (We use the standard notation $\tilde{\Omega}(n)$ to hide the dependence on poly-logarithmic factors. )

We consider the simple problem of finding the furthest vector for each vector in the given sequence of length $n$ and dimension $d \in \Omega(\log^2 n)$. The assumption on the dimension is mild , as in many situations the dimension $d = 768$ is actually comparable to the number of $n$.

**Task 1.** *Given $n$ unit vectors $\{u_1, \ldots, u_n\}$, each in $\mathbb{R}^d$ where $d = \Theta(\log^2 n)$, compute $f(u_1, \ldots, u_n) \rightarrow (u_{1^*}, \ldots, u_{n^*})$ where for a fixed $j \in [n]$, we define $j^* = \arg\max_k \|u_k - u_j\|_2^2$.*

Finding vectors that are furthest apart boils down to minimizing inner product search in case of unit vectors. For a full-attention mechanism with appropriate query and keys, this task is very easy as we can evaluate all pair-wise inner products.

The impossibility for sparse-attention follows from hardness results stemming from Orthogonal Vector Conjecture (OVC) [2, 1, 97, 7], which is a widely used assumption in fine-grained complexity. Informally, it states that one cannot determine if the minimum inner product among $n$ Boolean vectors is 0 in subquadratic time.

**Conjecture 1** (Orthogonal Vectors Conjecture). *For every $\epsilon > 0$, there is a $c \geq 1$ such that given $n$ Boolean vectors in $d$ dimension, cannot determine if there is a pair of orthogonal vectors in $O(n^{2-\epsilon})$ time on instances with $d \geq c \log n$.*

Using conjecture 1, we show a reduction to show that a transformer $g \in \mathcal{T}_D^{H=O(d), m=O(d), q=O(d)}$ for any sparse directed graph $D$ which completes Task 1 must require a superlinear number of layers.

**Proposition 2.** *There exists a single layer full-attention network $g \in \mathcal{T}^{H=1, m=2d, q=0}$ that can evaluate Task 1, i.e. $g(u_1, ..., u_n) = [u_{1^*}, \ldots, u_{n^*}]$, but for any sparse-attention network in $\mathcal{T}_D^{H=O(d), m=O(d), q=O(d)}$ with graph $D$ having $\tilde{O}(n)$ edges (i.e. inner product evaluations), would require $\tilde{\Omega}(n^{1-o(1)})$ layers.*

*Proof.* We will break this proof into two parts:

**Part 1: The full attention mechanism can solve the problem in $O(1)$ layer**   We begin by providing an explicit construction of a single layer full self-attention that can evaluate Task 1.

**Step 1** We embed each $u_i$ in the input into $\mathbb{R}^{2d}$ as follows:

$$x_i := E(u_i) = [u_i; 0] \tag{9}$$

**Step 2** Construct query, key, value functions as follows:

$$\begin{aligned} Q([a; b]) &= -a \\ K([a; b]) &= a \\ V([a; b]) &= [0; a] \end{aligned} \tag{10}$$

Then $\text{Attn}(Q(x_i), K(X), V(X) = [0; u_{\arg\max_j \langle -u_i, u_j \rangle}]$. Then,

$$a_i = \text{Attn}(Q(x_i), K(X), V(X)) + x_i = [u_i; u_{\arg\max_j \langle -u_i, u_j \rangle}] = [u_i; u_{i^*}] \tag{11}$$

**Step 3** Let $O(a_i) = 0$, then the output $z_i = [u_i; u_{i^*}]$ as desired.

To complete the argument, observe that it now only takes $O(n)$ inner products to check if there is a pair of orthogonal vectors as we need only compare $\langle u_i, u_{i^*} \rangle$.

**Part 2: Every Sparse Attention Mechanism will need $\tilde{\Omega}(n^{1-\epsilon})$ layers**    We prove by contradiction that it is impossible to solve Task 1 by any $g \in \mathcal{T}_D^{H=O(d),m=O(d),q=O(d)}$ sparse-attention graph $D$ with $\tilde{O}(n)$ edges.

Suppose we can solve Task 1 using a network $g \in \mathcal{T}_D^{H=O(d),m=O(d),q=O(d)}$ that has $l$ layers. Recall that all the computation we do in one layer is:

$$
\begin{aligned}
a_i &= \text{ATTN}_D(Q(x_i), K(X_{N(i)}), V(X_{N(i)})) + x_i \\
x_i &= O(a_i) + a_i
\end{aligned}
\tag{12}
$$

where $\text{Attn}_D$ is defined in eq. (AT).

Thus, total computation per layer is $\tilde{O}(nd^3)$ and consequently $\tilde{O}(nld^3)$ for the whole network consisting of $l$ layers.

We can use the result of Task 1 to solve the orthogonal vector (OV) problem (defined in Conjecture 1) in linear time. So in total, we will be able to solve any instance of OV in $\tilde{O}(nld^3)$ time.

Now if $l = O(n^{1-\epsilon})$ for any $\epsilon > 0$ and $d = \Theta(\log^2 n)$, then it appears that we are able to solve OV in $\tilde{O}(n^{2-\epsilon})$ which contradicts Conjecture 1. Therefore, we need at least $\tilde{\Omega}(n^{1-o(1)})$ layers.    $\square$

# D  Implementation details

We optimize the code for modern hardware. Hardware accelerators like GPUs and TPUs truly shine on coalesced memory operations which load blocks of contiguous bytes at once. Thus, its not very efficient to have small sporadic look-ups caused by a sliding window or random element queries. We alleviate this by "blockifying" the lookups.

**GPU/TPU and Sparsity**  Ideally, if the adjacency matrix $A$ described in Sec. 2 is sparse, one would hope this would be sufficient to speed up the implementation. Unfortunately, it is well known [33, 103], that such sparse multiplications cannot be efficiently implemented in GPUs. GPUs have thousands of cores performing operations in parallel. Thus, we cannot efficiently perform the sparse matrix multiplication mentioned in section Sec. 2.

As a result we propose to first blockify the attention pattern i.e. we pack sets of query and keys together and then define attention on these blocks. It is easier to explain this process using the example shown in Fig. 3. Suppose, there are 12 query and 12 key vectors to attend to. Using a block size of 2, we split the query matrix into $12/2 = 6$ blocks and similarly the key matrix into $12/2 = 6$ blocks. Then the three different building components of BIGBIRD are defined on the block matrix. In particular the three different components are:

1. Random attention: Each query block attends to $r$ random key blocks. In Fig. 3a, $r = 1$ with block size 2. This implies that each query block of size 2 randomly attends to a key block of size 2.

2. Window local attention: While creating the block, we ensure that the number of query blocks and the number of key blocks are the same. This helps us in defining the block window attention. Every query block with index $j$ attends to key block with index $j - (w-1)/2$ to $j + (w-1)/2$, including key block $j$. In Fig. 3b, $w = 3$ with block size 2. It means that each query block $j$ (size 2 queries) attends to key block $j - 1, j, j + 1$.

3. Global attention: Global attention remains the same as defined in Sec. 2, but we compute it in terms of blocks. In Fig. 3c, $g = 1$ with block size 2. For BIGBIRD-ITC this implies that one query and key block, attend to everyone.

The resulting overall attention matrix is shown in Fig. 3d. Unfortunately, simply trying to compute this attention score as multiplying arbitrary pairs of query and key vectors would require use of gather operation, which is inefficient. Upon closer examination of window and global attention, we observe that we can compute these attention scores without using a gather operation.

Recall, full dense attention scores can be calculated by simple matrix product of query and key matrix with a cost of $O(n^2d)$, as illustrated in Fig. 4a. Now note that if we blockify the query and key matrix and multiply, then with only $O(nbd)$ cost we will obtain the block diagonal portion of the attention score, as depicted in Fig. 4b. To elaborate this lets assume that $Q, K \in \mathbb{R}^{n \times d}$ are the query and key matrix corresponding to $n$ tokens such that $Q_{i.} = x_i W_Q$ and $K_{i.} = x_i W_K$. We reshape $n \times d$ query

| (a) Random Attention | (b) Window Attention | (c) Global Attention | (d) BIGBIRD |

Figure 3: Building blocks of the *block-attention* mechanism used in BIGBIRD with block size = 2. This implies the attention matrix is split into blocks of size $2 \times 2$. All the previous BIGBIRD parameters work on each block as a unit. White color indicates absence of attention. (a) random attention with $r = 1$, (b) sliding window attention with $w = 3$ (c) global attention with $g = 1$. (d) the combined BIGBIRD model.

(a) Full all pair attention can be obtained by direct matrix multiplication between the query and key matrix. Groupings just shown for guidance.

(b) Block diagonal attention can be computed by "blockifying" the query and key matrix

(c) Window local attention obtained by "blockifying" the query/key matrix, copying key matrix, and rolling the resulting key tensor (Obtaining rolled key-block tensor is illustrated in detail in Fig. 5). This ensures that every query attends to at least one block and at most two blocks of keys of size $b$ on each side.

(d) Window + Random attention obtained by following the procedure above along with gathering some random key blocks.

Figure 4: Idea behind fast sparse attention computation in BIGBIRD.

Figure 5: Construction of rolled key-block tensor. Make $w$ copies of the key matrix. Index the copies as $-(w-1)/2 \leq j \leq (w-1)/2$. Roll $j^{\text{th}}$ copy by $j$ blocks. Positive roll means circular shift entries left and likewise for negative roll corresponds to right shift. Finally, reshape by grouping the blocks along a new axis to obtain the key-blocked tensor. For illustration purpose $w = 3$ is chosen.

matrix, $Q$, and key matrix, $K$, along the sequence length to obtain $\lceil n/b \rceil \times b \times d$ tensors $Q'$ and $K'$ respectively. Now we multiply the two tensors as

$$A_{jst} = \sum_u Q'_{jsu} K'_{jtu}, \qquad j = 0, 1, ..., \lceil n/b \rceil \tag{13}$$

The resulting $A$ tensor of size $\lceil n/b \rceil \times b \times b$ can be reshaped to correspond to the block diagonal portion of the full attention pattern. Now to extend the attention from block diagonal to a window, i.e. where query block with index $j$ attends to key block with index $j - (w-1)/2$ to $j + (w-1)/2$, we make $w$ copies of the reshaped key tensor $K'$. We "roll" each copy of key-block tensor incrementally along the first axis of length $\lceil n/b \rceil$ as illustrated in Fig. 5. Multiplying these $w$ rolled key-block tensors with the query-block tensor would yield the desired window attention scores (Fig. 4c). Likewise the global component, we can always include the first $g$ blocks from key tensor corresponding to the global tokens. Finally, for the random attention, which is very small ($r = 3$ for all of our experiments), we resort to using gather ops (Fig. 4d). Also note by design, each query block attends to exactly $r$ random blocks.

Thus, the result of all the three components is basically a compact dense tensor $K''$ of size $\lceil n/b \rceil \times (g+w+r)b \times d$ as shown in Fig. 6. Computing the final attention score then just boils down to a dense tensor multiplication, at which TPU/GPU are very efficient. Specifically, we need to multiply $Q'$ (size: $\lceil n/b \rceil \times b \times d$) and $K''$ (size: $\lceil n/b \rceil \times (g+w+r)b \times d$) with a cost of $O(n(g+w+r)bd)$ to yield the desired attention score tensor of size $\lceil n/b \rceil \times b \times (g+w+r)b$, which can be reshaped to obtain all the attention scores according to the BigBird pattern.

Figure 6: Overview of BIGBIRD attention computation. Structured block sparsity helps in compactly packing our operations of sparse attention, thereby making our method efficient on GPU/TPU. On the left, we depict the transformed dense query and key tensors. The query tensor is obtained by simply blocking and reshaping while the final key tensor by concatenating three transformations: The first green columns, corresponding to global attention, is fixed. The middle blue columns correspond to window local attention and can be obtained by appropriately rolling as illustrated in Fig. 5. For the final orange columns, corresponding to random attentions, we need to use computationally inefficient gather operation. Dense multiplication between the query and key tensors efficiently calculates the sparse attention pattern (except the first row-block, which is computed by direct multiplication), using the ideas illustrated in Fig. 4. The resultant matrix on the right is same as that shown in Fig. 3d.

# E  NLP experiments details

## E.1  MLM Pretraining

We use four publicly available datasets Books [111], CC-News [34], Stories [90] and Wikipedia to pretrain BIGBIRD. We borrow the sentencepiece vocabulary as RoBERTa (which is in turn borrowed from GPT2). We split any document longer than 4096 into multiple documents and we join documents that were much smaller than 4096. Following the original BERT training, we mask 15% of tokens in these four datasets, and train to predict the mask. We warm start from RoBERTa's checkpoint. We train two different models: BIGBIRD-ITC-base and BIGBIRD-ETC-base. The hyper-parameters for these two models are given in Tab. 8. In all experiments we use a learning rate warmup over the first 10,000 steps, and linear decay of the learning rate.

Similar to the norm, we trained a large version of model as well, which has 24 layers with 16 heads and hidden dimension of 1024. Following the observation from RoBERTa, we pretrain on a larger batch size of 2048 for this size. For BIGBIRD-ITC the block length was kept same as base size, but for BIGBIRD-ETC the block length was almost doubled to 169. All the remaining parameters were the same.

Table 8: Hyperparameters for the two BIGBIRD base models for MLM.

| Parameter | BIGBIRD-ITC | BIGBIRD-ETC |
|---|---|---|
| Block length, $b$ | 64 | 84 |
| # of global token, $g$ | $2 \times b$ | 256 |
| Window length, $w$ | $3 \times b$ | $3 \times b$ |
| # of random token, $r$ | $3 \times b$ | 0 |
| Max. sequence length | 4096 | 4096 |
| # of heads | 12 | 12 |
| # of hidden layers | 12 | 12 |
| Hidden layer size | 768 | 768 |
| Batch size | 256 | 256 |
| Loss | MLM | MLM |
| Activation layer | gelu | gelu |
| Dropout prob | 0.1 | 0.1 |
| Attention dropout prob | 0.1 | 0.1 |
| Optimizer | Adam | Adam |
| Learning rate | $10^{-4}$ | $10^{-4}$ |
| Compute resources | $8 \times 8$ TPUv3 | $8 \times 8$ TPUv3 |

## E.2  Question Answering

The detailed statistics of the four datasets used are given in Tab. 11. All the hyperparameters for BIGBIRD, used for creating Tab. 2 are shown in Tab. 12 and those submitted to get Tab. 3 are shown in Tab. 13. We use two types of regularization in training:
- We used a variant of contrastive predictive coding [70] as a dual encoder model.
- We use position embedding for ITC and relative position encoding [79] for ETC.

Next, we will mention the dataset/task specific part of the model.

**HotpotQA**  The data consists of each question with multiple evidence paragraphs. We filtered 16 QA where the answer was not in the given evidences. For BIGBIRD-ITC, we use first 128 global tokens. For BIGBIRD-ETC, we have one global token for each question token, one for each evidence paragraph, and one for each sentence within the paragraph, for a maximum of 256 global token. We use a dense layer on the output corresponding to global token of the evidence paragraph to predict whether its a supporting fact with a threshold over the output logits. The answer type (yes/no/span) is

Table 9: Dataset used for pre training.

| Dataset | # tokens | Avg. doc len. |
|---|---|---|
| Books [111] | 1.0B | 37K |
| CC-News [34] | 7.4B | 561 |
| Stories [90] | 7.7B | 8.2K |
| Wikipedia | 3.1B | 592 |

Table 10: MLM performance on held-out set.

| Model | Base | Large |
|---|---|---|
| RoBERTa (sqln: 512) | 1.846 | 1.496 |
| Longformer (sqln: 4096) | 1.705 | 1.358 |
| BIGBIRD-ITC (sqln: 4096) | 1.678 | 1.456 |
| BIGBIRD-ETC (sqln: 4096) | **1.611** | **1.274** |

Table 11: Question Answering Datasets

| Dataset | Instances | | Instance Length | |
| --- | --- | --- | --- | --- |
| | Training | Dev | Median | Max |
| HotpotQA-distractor [101] | 90447 | 7405 | 1227 | 3560 |
| Natural Questions [52] | 307373 | 7830 | 3258 | 77962 |
| TriviaQA [41] | 61888 | 7993 | 4900 | 32755 |
| WikiHop [96] | 43738 | 5129 | 1541 | 20337 |

Table 12: Hyperparameters of base BIGBIRD model used for Question Answering i.e. the numbers reported in Tab. 2

| Parameter | HotpotQA | | NaturalQ | | TriviaQA | | WikiHop | |
| --- | --- | --- | --- | --- | --- | --- | --- | --- |
| Global token location | ITC | ETC | ITC | ETC | ITC | ETC | ITC | ETC |
| # of global token, $g$ | 128 | 256 | 128 | 230 | 128 | 320 | 128 | 430 |
| Window length, $w$ | 192 | 252 | 192 | 252 | 192 | 252 | 192 | 252 |
| # of random token, $r$ | 192 | 0 | 192 | 0 | 192 | 0 | 192 | 0 |
| Max. sequence length | 4096 | 4096 | 4096 | 4096 | 4096 | 4096 | 4096 | 4096 |
| # of heads | 12 | 12 | 12 | 12 | 12 | 12 | 12 | 12 |
| # of hidden layers | 12 | 12 | 12 | 12 | 12 | 12 | 12 | 12 |
| Hidden layer size | 768 | 768 | 768 | 768 | 768 | 768 | 768 | 768 |
| Batch size | 32 | 32 | 128 | 128 | 32 | 32 | 64 | 64 |
| Loss | cross-entropy golden spans | | cross-entropy golden spans | | cross-entropy noisy spans [18] | | cross-entropy ans choices | |
| Compute resources | $4 \times 2$ TPUv3 | | $4 \times 8$ TPUv3 | | $4 \times 2$ TPUv3 | | $4 \times 4$ TPUv3 | |

predicted with a single dense layer from the global CLS token. For span based answers, the spans are predicted with dense layers on the sequence with the distance between start and end positions to be no more than 30 words. The spans are ranked by sum of start and end logits.

**Natural Questions**   Here also the data consists of question with supporting evidence, but in form of a single, potentially long, document and not multiple paragraphs. We largely follow the setup of [5]. For documents, that are longer than 4096, a sliding window approach is used with stride of 2048. We use CLS token at the beginning, followed by the question followed by a separator token followed by the document as input. For BIGBIRD-ITC, we make the first 128 tokens as global. For BIGBIRD-ETC, we make a global token for CLS, question, and one token for each of the paragraphs. We train four predictors at the final layer to predict long answer start, long answer end, short answer start and short answer end respectively. Instead of independently predicting the start and end of answers we first predict the start and then predict the best end location beyond the start. For short answer, we limit the distance between start and end positions to be no more than 38 words. The answer type (null, yes, no, short, long) is predicted from CLS token output embedding. When the logit for a yes/no answer is higher than the logits for short, long or null answer, we replace the short answer with a corresponding yes/no text.

**TriviaQA**   The data consists of question-answer pairs with Wikipedia articles as the "noisy" supporting evidence. We call them noisy because the given Wikipedia articles may or may not contain the answer. Moreover, the answer entities is not annotated to appropriate span in the article, rather all occurrences found using fuzzy string matching are listed. We use CLS token at the beginning, followed by the question followed by a separator token followed by the document as input. For BIGBIRD-ITC, we make the first 128 tokens as global. For BIGBIRD-ETC, we make a global token for CLS, question, and one token for each sentence up to a maximum of 320 global tokens. Given the noisy nature of answer span, we follow Clark and Gardner [18] for training. We use a dense layer on the sequence to predict the answer span for each article independently, with the distance between start and end positions to be no more than 16 words. For each article the span with maximum start logit + end logit is chosen. Then we normalize over all the documents associated with that question.

**WikiHop**   For each question in WikiHop, we are given upto 79 candidates, and 63 supporting paragraphs. In our BIGBIRD-ITC model, following Beltagy et al. [8], we concatenate the answer and

Table 13: Hyperparameters of large BIGBIRD model for Question Answering submitted for test i.e. the numbers reported in Tab. 3

| Parameter | HotpotQA | NaturalQ | TriviaQA | WikiHop |
|---|---|---|---|---|
| Global token location | ETC | ETC | ETC | ETC |
| # of global token, $g$ | 256 | 230 | 320 | 430 |
| Window length, $w$ | 507 | 507 | 507 | 507 |
| # of random token, $r$ | 0 | 0 | 0 | 0 |
| Max. sequence length | 4096 | 4096 | 4096 | 4096 |
| # of heads | 16 | 16 | 16 | 16 |
| # of hidden layers | 24 | 24 | 24 | 24 |
| Hidden layer size | 1024 | 1024 | 1024 | 1024 |
| Batch size | 32 | 64 | 32 | 64 |
| Loss | cross-entropy | cross-entropy | cross-entropy | cross-entropy |
| Num epochs | $\{5, 9\}$ | $\{3, 5\}$ | $\{3, 5\}$ | $\{5, 10\}$ |
| Optimizer | Adam | Adam | Adam | LAMB |
| Learning rate | $3 \times 10^{-5}$ | $\{5, 10\} \times 10^{-5}$ | $\{3, 5\} \times 10^{-5}$ | $\{2, 5\} \times 10^{-5}$ |
| Compute resources | $4 \times 4$ TPUv3 | $4 \times 8$ TPUv3 | $4 \times 4$ TPUv3 | $4 \times 8$ TPUv3 |

the question with special tokens, `[q] Question [/q] [ans] Ans1 [/ans] ... [ans] AnsN [/ans]` along with the context. As the start of the text, always contains questions followed by answers, we make the first 128 token attend globally. In BIGBIRD-ETC model, we do not need to insert special `[ans]`, `[/ans]` etc. as we design global tokens appropriately. Along with global tokens for question, we have one per candidate answer up to a maximum of 430. Further, we linked answer tokens to their mentions using relative position label. Lastly, we use a dense layer that takes in the output vector corresponding to a candidate answer, and predicts a score for the current candidate to be the correct answer. We apply this dense layer to each candidate independently and the candidate with the best score is picked as our final answer.

It is worthwhile to note that explicitly designed attention connection in ETC works slightly better, the random connection based ITC is pretty competative.

### E.3 Relationship to Contemporary Work

**Longformer** Child et al. [16] introduced localized sliding window to reduce computation. A more recent version, which includes localized sliding windows and global tokens was introduced independently by Longofrmer[8]. Although BIGBIRD contains additional random tokens, there are also differences in the way global and local tokens are realized. In particular even when there is no random token, as used to get SoTA in question answering, there are two key differences between Longformer and BIGBIRD-etc (see [4]):

1. We use global-local attention with relative position encodings enables it to better handle structured inputs
2. Unlike Longformer, we train the global tokens using CPC loss and learn their use during finetuning.

### E.4 Classification

We try two types of classification task.

**Document classification** We experiment on datasets of different lengths and contents, as listed in Tab. 15. In particular, we look at sentiment analysis (IMDb [64] and Yelp-5 [109]) task and topic assignment (Arxiv [35], Patents [53], and Hyperpartisan [47]) task. Following BERT, we used one layer with cross entropy loss on top of the first [CLS] token from the BIGBIRD encoder consuming 4096 tokens. We report the results of document classification experiments in Tab. 15. We compare against state-of-the-art (SoTA) methods for each dataset and plain RoBERTa model with 512 tokens truncation. In all experiments we use a learning rate warmup over the first 10% steps, and linear decay of the learning rate and detail list of remaining hyperparameters are provided in Tab. 14. For better quantitative evaluation, we compute the fraction of the dataset that exceeds 512 tokens, i.e. the length at which the document are often truncated. We see that gains of using BIGBIRD are more significant when we have longer documents and fewer training examples. For instance, using base sized model,

Table 14: Hyperparameters for document classification.

| Parameter | IMDb | Arxiv | Patents | Hyperpartisan | Yelp-5 |
|---|---|---|---|---|---|
| Batch size | 64 | 64 | 64 | 32 | 32 |
| Learning rate | $1 \times 10^{-5}$ | $3 \times 10^{-5}$ | $5 \times 10^{-5}$ | $5 \times 10^{-6}$ | $2 \times 10^{-5}$ |
| Num epochs | 40 | 10 | 3 | 15 | 2 |
| TPUv3 slice | $4 \times 4$ | $4 \times 4$ | $4 \times 4$ | $4 \times 2$ | $4 \times 8$ |
| # of heads | | | 12 | | 16 |
| # of hidden layers | | | 12 | | 24 |
| Hidden layer size | | | 768 | | 1024 |
| Block length, $b$ | | | 64 | | |
| Global token location | | | ITC | | |
| # of global token, $g$ | | | $2 \times b$ | | |
| Window length, $w$ | | | $3 \times b$ | | |
| # of random token, $r$ | | | $3 \times b$ | | |
| Max. sequence length | | | 4096 | | |
| Vocab size | | | 50358 | | |
| Activation layer | | | gelu | | |
| Dropout prob | | | 0.1 | | |
| Attention dropout prob | | | 0.1 | | |
| Loss | | | cross-entropy | | |
| Optimizer | | | Adam | | |

Table 15: Classification results. We report the F1 micro-averaged score for all datasets. Experiments on smaller IMDb and Hyperpartisan datasets are repeated 5 times and the average performance is presented along with standard deviation.

| Model | IMDb [64] | Yelp-5 [109] | Arxiv [35] | Patents [53] | Hyperpartisan [47] |
|---|---|---|---|---|---|
| # Examples | 25000 | 650000 | 30043 | 1890093 | 645 |
| # Classes | 2 | 5 | 11 | 663 | 2 |
| Excess fraction | 0.14 | 0.04 | 1.00 | 0.90 | 0.53 |
| SoTA | [89] 97.4 | [3] 73.28 | [69] 87.96 | [69] 69.01 | [40] 90.6 |
| RoBERTa | $95.0 \pm 0.2$ | 71.75 | 87.42 | 67.07 | $87.8 \pm 0.8$ |
| BIGBIRD | $95.2 \pm 0.2$ | 72.16 | **92.31** | 69.30 | $\mathbf{92.2 \pm 1.7}$ |

BIGBIRD improves state-of-the-art for Arxiv dataset by about **5% points**. On Patents dataset, there is improvement over using simple BERT/RoBERTa, but given the large size of training data the improvement over SoTA (which is not BERT based) is not significant. Note that this performance gain is not seen for much smaller IMDb dataset. Along with experimental setup detail, we present detailed results in App. E.4 which show competitive performance.

**GLUE** The General Language Understanding Evaluation (GLUE) benchmark [93], test language models on 8 different natural language understanding tasks. We used the same training parameters as mentioned in `https://github.com/pytorch/fairseq/blob/master/examples/roberta/README.glue.md`. Our model parameters are $b = 64, g = 2 \times b, w = 3 \times b, r = 3 \times b$ ( we used the BIGBIRD-ITC base model pretrained on MLM task). We compare the performance of BIGBIRD to BERT, XLNet [102] and RoBERTa in Tab. 16. We find that even on task that have a much smaller context, our performance is competitive to full attention models.

Table 16: GLUE Dev results on base sized models. Number of training examples is reported below each task. MCC score is reported for CoLA, F1 score is reported for MRPC, Spearman correlation is reported for STS-B, and accuracy scores are reported for the other tasks.

| System | MNLI-(m/mm) 392k | QQP 363k | QNLI 108k | SST-2 67k | CoLA 8.5k | STS-B 5.7k | MRPC 3.5k | RTE 2.5k |
|---|---|---|---|---|---|---|---|---|
| BERT | 84.6/83.4 | 71.2 | 90.5 | 93.5 | 52.1 | 85.8 | 88.9 | 66.4 |
| XLNet | 86.8/- | 91.4 | 91.7 | 94.7 | 60.2 | 89.5 | 88.2 | 74.0 |
| RoBERTa | 87.6/- | 91.9 | 92.8 | 94.8 | 63.6 | 91.2 | 90.2 | 78.7 |
| BIGBIRD | 87.5/87.3 | 88.6 | 92.2 | 94.6 | 58.5 | 87.8 | 91.5 | 75.0 |

## E.5 Summarization

As discussed in Sec. 4.1, given the small length of output sequence, we used sparse BIGBIRD attention only for encoder, while keeping the full attention for decoder. The number of hidden layers, number of heads, and hidden dimension is same for encoder and decoder. The hyperparameters are detailed in Tab. 17. We summarize our result in Tab. 20. In all experiments, we use a learning rate warmup over the first 10,000 steps, and square root decay of the learning rate.

Table 17: Encoder hyperparameters for Summarization. We use full attention in decoder

| Parameter | | Base: BIGBIRD-RoBERTa | Large: BIGBIRD-Pegasus |
|---|---|---|---|
| Block length, $b$ | | 64 | 64 |
| Global token location | | ITC | ITC |
| # of global token, $g$ | | $2 \times b$ | $2 \times b$ |
| Window length, $w$ | | $3 \times b$ | $3 \times b$ |
| # of random token, $r$ | | $3 \times b$ | $3 \times b$ |
| Max. encoder sequence length | BBC-XSUM: | 1024 | 1024 |
| | CNN/DM: | 2048 | 2048 |
| | Others: | 3072 | 3072 |
| Max. decoder sequence length | BBC-XSUM: | 64 | 64 |
| | CNN/DM: | 128 | 128 |
| | Others: | 256 | 256 |
| Beam size | | 5 | 5 |
| Length penalty | BBC-XSUM: | 0.7 | 0.7 |
| | Others: | 0.8 | 0.8 |
| # of heads | | 12 | 16 |
| # of hidden layers | | 12 | 16 |
| Hidden layer size | | 768 | 1024 |
| Batch size | | 128 | 128 |
| Loss | | teacher forced cross-entropy | teacher forced cross-entropy |
| Activation layer | | gelu | gelu |
| Dropout prob | | 0.1 | 0.1 |
| Attention dropout prob | | 0.1 | 0.1 |
| Optimizer | | Adam | Adafactor |
| Learning rate | | $1 \times 10^{-5}$ | $1 \times 10^{-4}$ |
| Compute resources | | $4 \times 4$ TPUv3 | $4 \times 8$ TPUv3 |

Table 18: Statistics of datasets used for summarization.

| Dataset | Instances | | | Input Length | | Output Length | |
|---|---|---|---|---|---|---|---|
| | Training | Dev | Test | Median | 90%-ile | Median | 90%-ile |
| Arxiv [20] | 203037 | 6436 | 6440 | 6151 | 14405 | 171 | 352 |
| PubMed [20] | 119924 | 6633 | 6658 | 2715 | 6101 | 212 | 318 |
| BigPatent [78] | 1207222 | 67068 | 67072 | 3082 | 7693 | 123 | 197 |

Following success of several recent works [76, 63], we warm start our encoder-decoder BIGBIRD transformer model with pretrained weights and the weights between encoder and decoder are shared. In particular, the query/key/value matrix of self-attention and all the feedforward layers are shared between encoder and decoder. The only variable that is initialized randomly is the encoder-decoder attention. For base sized model, we utilize our MLM pretrained model on 4096 sequence length from App. E.1, which is in turn initialized using the public RoBERTa checkpoint. For the large size model, we lift weight from the state-of-the-art Pegasus model [108], which is pretrained using an objective designed for summarization task.

To check if sparse attention causes significant degradation as compared to full attention, we further experiment on two shorter but popular datasets, where full attention can be used without significantly truncating the document. The statistics of these two datasets are in Tab. 19. We see that our performance is competitive, which shows that sparse attention can achieve similar performance to a full attention models.

Table 19: Shorter summarization dataset statistics.

| Dataset | Instances | | | Input Length | | Output Length | |
|---|---|---|---|---|---|---|---|
| | Training | Dev | Test | Median | 90%-ile | Median | 90%-ile |
| BBC XSum [67] | 204044 | 11332 | 11334 | 359 | 920 | 25 | 32 |
| CNN/DailyMail [36] | 287113 | 13368 | 11490 | 777 | 1439 | 59 | 93 |

Table 20: Summarization ROUGE score for shorter documents.

| Model | | BBC XSum | | | CNN/DailyMail | | |
|---|---|---|---|---|---|---|---|
| | | R-1 | R-2 | R-L | R1 | R2 | R-L |
| Prior Art | Lead | 16.30 | 1.61 | 11.95 | 39.60 | 17.70 | 36.20 |
| | PtGen [77] | 29.70 | 9.21 | 23.24 | 39.53 | 17.28 | 36.38 |
| | ConvS2S [28] | 31.89 | 11.54 | 25.75 | – | – | – |
| | MMN [48] | 32.00 | 12.10 | 26.00 | – | – | – |
| | Bottom-Up [29] | – | – | – | 41.22 | 18.68 | 38.34 |
| | TransLM [45] | – | – | – | 39.65 | 17.74 | 36.85 |
| | UniLM [23] | – | – | – | 43.47 | 20.30 | 40.63 |
| | Extr-Abst-BERT [62] | 38.81 | 16.50 | 31.27 | 42.13 | 19.60 | 39.18 |
| | BART [56] | 45.14 | 22.27 | 37.25 | 44.16 | 21.28 | 40.90 |
| Base | Transformer [92] | 29.61 | 9.47 | 23.17 | 34.89 | 13.13 | 32.12 |
| | + RoBERTa [76] | 39.92 | 17.33 | 32.63 | 39.44 | 18.69 | 36.80 |
| | + Pegasus [108] | 39.79 | 16.58 | 31.70 | 41.79 | 18.81 | 38.93 |
| | BIGBIRD-RoBERTa | 39.52 | 17.22 | 32.30 | 39.25 | 18.46 | 36.61 |
| Large | Pegasus (Reported) [108] | 47.60 | 24.83 | 39.64 | 44.16 | 21.56 | 41.30 |
| | Pegasus (Re-eval) | **47.37** | **24.31** | **39.23** | **44.15** | **21.56** | **41.05** |
| | BIGBIRD-Pegasus | 47.12 | 24.05 | 38.80 | 43.84 | 21.11 | 40.74 |

# F   Genomics experiments details

In this section we provide details of the experimental setup for BIGBIRD on genomics data.

## F.1   Pretraining

We try to keep the experimental setup as close to a typical NLP pipeline. In this regard, we take human reference GRCh37[7] and convert it into documents $\mathcal{D}$. Each document $d \in \mathcal{D}$ is a sequence of sentences, where each sentence is a sequence of fragments of DNA. We construct the documents as follows:

1. Start with empty document set $D = \emptyset$.
2. For each chromosome $C$, repeat the following procedure 10 times.
   (a) Pick uniformly at random a starting point $q$ between base pairs 0 and 5000 from the 5' end.
   (b) Repeat until $q > |C|$
      i. Pick uniformly at random $s$ a number between 50 and 100 to denote number of sentences per document.
      ii. Constructs a document $d$ containing $s$ sentences using consecutive base pairs (bps). The length of each sentence is chosen uniformly at random between 500-1000. Thus the resulting document has $25,000$ - $100,000$ bps.
      iii. $D = D \bigcup d$
      iv. $q = q + |d|$

By this procedure we end-up with approximately $450K$ documents.

Next we run sentencepiece [50] tokenization on the resulting documents. In particular, using 5 characters as the building blocks (four for bases - A, T, C, G and one for missing symbol N), we

Figure 7: Visual description of how the masked language modeling data was generated from raw DNA dataset. The raw DNA sequences of GRCh37, where split at random positions to create documents with 50-100 sentences where each sentence was 500-1000 base pairs (bps). Thus each document had a continuous strand of 25000-100,000 bps of DNA. This process was repeated 10 times to create 10 sets of document for each chromosome of GRCH37. The resulting set of documents was then passed through Sentencepiece that created tokens of average 8bp. For pretraining we used masked language model and masked $10\%$ of the tokens and trained on predicting the masked tokens.

construct a byte pair encoding table of size 32k, with each token representing 8.78 base pairs on average.

Using the above constructed documents, we construct a dataset for two pretraining tasks following Devlin et al. [22]:

- **Masked Language Model (MLM):** In order to train a deep bidirectional representation, BERT training introduces the MLM task, where we simply mask out 15% of the input tokens at random, and then predict those masked tokens. We can simply replace such masked out of the tokens with a [MASK] placeholder, but it leads to a distribution mis-match for downstream tasks which will not have such placeholders. To mitigate with this issue, out of the 15% of the tokens selected for masking:

  - 80% of the tokens are actually replaced with the token [MASK].
  - 10% of the time tokens are replaced with a random token.
  - 10% of the time tokens are left unchanged, but are still predicted at output.

  We run this entire sequence through the BIGBIRD transformer encoder and then predict corresponding to the masked positions, based on the context provided by the other non-masked tokens in the sequence.

- **Next Sentence Prediction (NSP):** In order to understand relationship between two sequences, BERT training introduces the NSP task, where we predict if a given pair of sequences are contiguous or not. During training the model gets as input pairs of sequences separated by [SEP] token along with a [CLS] token at the start. Overall the input pattern

Figure 9: Visual description of the DNA segment from which we predict the chromatin profile for a given non-coding region of the raw DNA sequences of GRCh37. We take 8000 bps of DNA before and after the given non-coding region as context. The complete fragment of DNA including the context on both side, is then tokenized to form our input sequence of tokens. The task is to predict 919 chromatin profile including 690 transcription factors (TF) binding profiles for 160 different TFs, 125 DNase I sensitivity (DHS) profiles and 104 histone-mark (HM) profiles

is: [CLS] sequence A [SEP] sequence B [SEP]. For 50% of the time the second sequence comes from true sequence after the first one. Remaining 50% of the time it is a a random sequence from the full dataset. The model is then required to predict this relationship using the output corresponding to the [CLS] token, which is fed into a simple binary classification layer.

The sequence of steps is visually elaborated in Fig. 9. The model is trained with both MLM and NSP together. Training hyperparameter is provided in second columns of Tab. 21. In all experiments we use a learning rate warmup over the first 10,000 steps, and linear decay of the learning rate.

We additionally performed a simple ablation study to validate the hypothesis, that similar to NLP, having a larger context improves performance. We use MLM task described above to test how BIG-BIRD performed with sequences of different length. Accuracy on MLM task with increasing sequence length is shown in Fig. 8. Not only longer context improves final accuracy, it also leads to faster learning, as we have now more opportunities for masking.

Figure 8: BIGBIRD accuracy with context length.

## F.2 Promoter Region Prediction

The promoter region plays an important role in transcription initiation and thus its recognition is an important area of interest in the field of bioinformatics. Following Oubounyt et al. [71], we use datasets from Eukaryotic Promoter Database (EPDnew) [24], which contains 29,597 promoter region in the human genome. Around the transcription start site (TSS), we extract a sequence of 8000 bp (-5000 +3000 bp) from the human reference genome GRCh37. Since EPDnew uses newer GRCh38, we convert to GRCh37 coordinates using LiftOver [44].

Following Oubounyt et al. [71] for each promoter region example, a negative example (non-promoter sequences) with the same size of the positive one is constructed as follow: The positive sequence is divided into 20 subsequences. Then, 12 subsequences are picked randomly and substituted randomly. The remaining 8 subsequences are conserved. This process is illustrated in Figure 1 of [71]. Applying this process to the positive set results in new non-promoter sequences with conserved parts from promoter sequences (the unchanged subsequences, 8 subsequences out of 20). These parameters enable generating a negative set that has 32 and 40% of its sequences containing conserved portions of promoter sequences.

We prefix and append each example with [CLS] and [SEP] token respectively. The output corresponding to the [CLS] token from BIGBIRD transformer encoder is fed to a simple binary classification layer. We fine-tune the pretrained BIGBIRD from App. F.1 using hyper-parameters described in Tab. 21.

Table 21: Table of hyperparameters for Computational biology.

| Parameter | Pretraining | Promoter Region | Chromatin-Profile |
|---|---|---|---|
| Block length, $b$ | 64 | 64 | 64 |
| Global token location | ITC | ITC | ITC |
| # of global token, $g$ | $2 \times b$ | $2 \times b$ | $2 \times b$ |
| Window length, $w$ | $3 \times b$ | $3 \times b$ | $3 \times b$ |
| # of random token, $r$ | $3 \times b$ | $3 \times b$ | $3 \times b$ |
| Max. Sequence Length | 4096 | 4096 | 4096 |
| # of heads | 12 | 12 | 12 |
| # of hidden layers | 12 | 12 | 12 |
| Hidden layer size | 768 | 768 | 768 |
| Batch Size | 256 | 256 | 256 |
| Vocab Size | 32000 | 32000 | 32000 |
| Loss | MLM+NSP | BCE | 919 x +ve upweighted BCE |
| Dropout prob | 0.1 | 0.1 | 0.1 |
| Optimizer | Adam | Adam | Adam |
| Learning rate | 0.0001 | 0.0001 | 0.0001 |
| # of steps | 1000000 | 711 | 500000 |
| Compute Resources | $8 \times 8$ TPUv3 | $8 \times 8$ TPUv3 | $8 \times 8$ TPUv3 |

We note that high performance is not surprising due to the overlap in the nature of negative example generation and MLM pretraining.

### F.3 Chromatin-Profile Prediction

The first step of sequence-based algorithmic framework for predicting non-coding effects is to build a model to predict, large scale chromatic profile [110]. In this paper, we use the dataset provided in Zhou and Troyanskaya [110][8], to train BIGBIRD to predict the chromatic profile.

Each training sample consists of a 8,000-bp sequence from the human GRCh37 reference genome centered on each 200-bp bin and is paired with a label vector for 919 chromatin features. As before, we prefix and append each example with [CLS] and [SEP] token respectively. The output corresponding to the [CLS] token from BIGBIRD transformer encoder is fed to a linear layer with 919 heads. Thus we jointly predict the 919 independent binary classification problems. We fine-tune the pretrained BIGBIRD from App. F.1 using hyper-parameters described in Tab. 21. As the data is highly imbalanced data (way more negative examples than positive examples), we upweighted loss function for positive examples by factor of 8.

We used training and testing split provided by Zhou and Troyanskaya [110] using chromosomes and strictly non-overlapping. Chromosome 8 and 9 were excluded from training to test chromatin feature prediction performances, and the rest of the autosomes were used for training and validation. 4,000 samples on chromosome 7 spanning the genomic coordinates 30,508,751–35,296,850 were used as the validation set.

As the predicted probability for each sequence in DeepSea Zhou and Troyanskaya [110] was computed as the ensemble average of the probability predictions for the forward and complementary sequence pairs, we also predict using an ensemble of two BIGBIRD model trained independently.

## Footnotes

[7] https://www.ncbi.nlm.nih.gov/assembly/GCF_000001405.39

[8]http://deepsea.princeton.edu/media/code/deepsea_train_bundle.v0.9.tar.gz