[Reviews · NeurIPS 2020]

Review 1

Summary and Contributions: The Big Bert model proposed in this paper contains sparse attention that can handle long sequences without increasing the requirement of hardware. The results on various NLP tasks support their method.

Strengths: - Modeling the long-range of text for the BERT-based model is challenging. The sparse attention proposed in this paper is quite interesting, especially, they claim that it can handle sequences of length up to 8x of previous works. - They also provide theoretical results to support their sparse attention. - The experiments on QA and document classification looks quite good (compared with the state-of-the-art methods)

Weaknesses: While I quite agree that modeling long text is challenging for current Transformer (Vaswani). One of the inspirations of this work is "locality of reference" which assumes that a token can be derived from its neighboring tokens. However, for a document with a series of paragraphs, sometimes, the token may be related to the sentence in another paragraph. I think this is the weakness of the sliding window in BigBERT. Did the authors conduct the speed (inference time) comparison between their BigBERT and other methods?

Correctness: The method and experiment of this work are solid and convincing.

Clarity: This paper is well-written.

Relation to Prior Work: This paper provides a solid comparison with other methods.

Reproducibility: Yes

Additional Feedback: The feedback resolve my questions, I keep my suggestion.


Review 2

Summary and Contributions: The authors pointed that the full self-attention have computational and memory requirement that is quadratic in the sequence length. They produce a sparse attention mechanism that improves performance on a multitude of tasks requiring long contexts. At meanwhile, they proved the proposed BIGBIRD is a universal approximator of sequence functions and is Turing complete, thereby preserving these properties of the quadratic, full attention model.

Strengths: The overall motivation and theoretical part of the article is detailed and clear. The proposed sparse attention mechanism improves performance on a multitude of tasks that require long contexts and satisfies all the known theoretical properties of full transformer. Moreover, the experiments for proving the performance of the model is much more important are also sufficient.

Weaknesses: The authors select three representative NLP tasks to showcase benefits of modeling longer input sequence. However, there is no experiment to show whether the model performs well in the short text than other models.

Correctness: The claims, method and empirical methodology are all correct.

Clarity: This paper is well organized and clearly described.

Relation to Prior Work: yes

Reproducibility: Yes

Additional Feedback: It’s a good job. This paper is well organized and clearly described, whose overall motivation and theoretical part of the article is detailed. But, I suggest that the experiment section can be enriched, like I said above.


Review 3

Summary and Contributions: The authors propose a sparse attention mechanism that reduces this quadratic dependency issue in self-attention. The model consists of three types of attention mechanism: global attention on fixed positions, local attention in sliding-window and attention on random positions. The authors prove that the proposed method can preserve the properties of full attention model and achieve SOTA on a variety of NLP tasks.

Strengths: 1. The authors provide theoretical analysis on sparse attention mechanism which is interesting and useful for further work in this direction. 2. It's interesting to see random attention only can also work on SQuAD and MNLI tasks. 3. The experiment results are quite solid. The proposed model achieves SOTA on a variety of NLP tasks which cover multi-hop QA, QA with longer context and document classification. It is also tested on genomics data.

Weaknesses: 1. Although the random attention in BigBird is interesting, the global attention and local attention in sliding window are not novel, similar to sparse-transformer (Generating Long Sequences with Sparse Transformers). 2. The model doesn't work well on short answer extraction of natural question dataset.

Correctness: Yes

Clarity: Yes

Relation to Prior Work: `Yes

Reproducibility: Yes

Additional Feedback: I don't agree "reduces this quadratic dependency to linear" in line 4. It seems not linear and relies on the window size. Please clarify this. Typo: line 155, "turning complete"

[Author Response · NeurIPS 2020]

We thank the reviewers for their perceptive and useful comments. Subsequent to our submission, we expanded our
experiments to include more encoder-only tasks and more seq2seq generative task like summarization. As reported
in Tab. 1 and Tab. 2, we found that **BIGBIRD achieves new state-of-the-art (SoTA) for summarization task**. Also,
we **obtained TriviaQA results to establish a new SoTA with 84.50 F1 on full and 92.39 F1** on verified subset.
Furthermore for classification BIGBIRD achieves better performance than BERT. We will include these expanded results
and full details of the experimental setup/hyper-parameters in the final version of the paper with the extra page.

| Model | | Arxiv | | | PubMed | | | BigPatent | | |
|---|---|---|---|---|---|---|---|---|---|---|
| | | R-1 | R-2 | R-L | R-1 | R-2 | R-L | R-1 | R-2 | R-L |
| *Prior Art* | Attn-Seq2Seq | 29.30 | 6.00 | 25.56 | 31.55 | 8.52 | 27.38 | 28.74 | 7.87 | 24.66 |
| | Pntr-Gen-Seq2Seq | 32.06 | 9.04 | 25.16 | 35.86 | 10.22 | 29.69 | 33.14 | 11.63 | 28.55 |
| | Long-Doc-Seq2Seq | 35.80 | 11.05 | 31.80 | 38.93 | 15.37 | 35.21 | - | - | - |
| | Sent-CLF | 34.01 | 8.71 | 30.41 | 45.01 | 19.91 | 41.16 | 36.20 | 10.99 | 31.83 |
| | Sent-PTR | 42.32 | 15.63 | 38.06 | 43.30 | 17.92 | 39.47 | 34.21 | 10.78 | 30.07 |
| | Extr-Abst-TLM | 41.62 | 14.69 | 38.03 | 42.13 | 16.27 | 39.21 | 38.65 | 12.31 | 34.09 |
| | Dancer | 42.70 | 16.54 | 38.44 | 44.09 | 17.69 | 40.27 | - | - | - |
| *Base* | Transformer | 28.52 | 6.70 | 25.58 | 31.71 | 8.32 | 29.42 | 39.66 | 20.94 | 31.20 |
| | + RoBERTa | 31.98 | 8.13 | 29.53 | 35.77 | 13.85 | 33.32 | 41.11 | 22.10 | 32.58 |
| | + Pegasus | 34.81 | 10.16 | 30.14 | 39.98 | 15.15 | 35.89 | 43.55 | 20.43 | 31.80 |
| | BIGBIRD-RoBERTa | 41.22 | 16.43 | 36.96 | 43.70 | 19.32 | 39.99 | 55.69 | 37.27 | 45.56 |
| *Large* | Pegasus (Reported) | 44.21 | 16.95 | 38.83 | 45.97 | 20.15 | 41.34 | 52.29 | 33.08 | 41.75 |
| | Pegasus (Re-eval) | 43.85 | 16.83 | 39.17 | 44.53 | 19.30 | 40.70 | 52.25 | 33.04 | 41.80 |
| | BIGBIRD-Pegasus | **46.63** | **19.02** | **41.77** | **46.32** | **20.65** | **42.33** | **60.64** | **42.46** | **50.01** |

Table 1: Summarization ROUGE score for long documents.

| Model | IMDb | Yelp-5 | Arxiv | Patents | Hyperpartisan |
|---|---|---|---|---|---|
| SoTA | 97.4 | 73.28 | 87.96 | 69.01 | 90.6 |
| RoBERTa | $95.0 \pm 0.2$ | 71.75 | 87.42 | 67.07 | $87.8 \pm 0.8$ |
| BIGBIRD | $95.2 \pm 0.2$ | 72.16 | **92.31** | 69.30 | $\mathbf{92.2 \pm 1.7}$ |

Table 2: Classification results. We report the F1 micro-averaged score for all datasets.

Next we will answer the specific questions asked by each reviewers.

**R2, Modeling relationship between tokens in different paragraphs:** We agree that window attention models
"locality of reference", but global attention captures long distance relationships. BIGBIRD also uses random attention
which is motivated from the ability of random graphs to capture properties of fully connected graphs, hence adding
another way to capture long distance relationships. Moreover, our theoretical analysis shows that BIGBIRD is able to
capture all sequence to sequence functions, including the ones that have long range dependency, while our empirical
results back this claim by outperforming baselines.

**R2, Inference Time:** We compared BIGBIRD and BERT on sequences of length 512 and 1024 and found the inference
time to be comparable. BERT uses the full attention mechanism and thus goes out of memory for sequence with more
than 1K tokens. We will include this in the paper.

**R3, Experiments on shorter text:** Results from experiments on shorter text have been reported in table 15, section E.4
in appendix. The table compares performance of BIGBIRD on 8 different General Language Understanding Evaluation
(GLUE) benchmark tasks. We see that BIGBIRD performs competitively even on smaller input sequences.

**R4, Related work:** While window attention models have been proposed before, prior models were based on heuristics
and were not as versatile and robust as the original transformer. In particular, the same architecture did not attain SoTA
on multiple standard benchmarks nor handle both encoding and decoding. Moreover, these approximations did not
come with any theoretical guarantees. We both extend the theoretical understanding of sparse models and provide
BIGBIRD-attention architecture that achieves SoTA for multiple applications.

**R4, Quadratic to Linear:** We made an asymptotic statement assuming window size is constant as sequence length
grows. In particular, for $N$ tokens the total number of attention in BIGBIRD is upper bounded $N(2b + w + r)$ instead
of $N^2$ in BERT and transformers. Here $b, w, r$ are the size of global attention, local attention and random attention per
query respectively. These sizes are kept constant for all the experiments leading to attention being linear in the number
of token. We have reported results when $N = 4096$, but have conducted experiments with $N > 16,000$ tokens, where
$N \gg 2b + w + r$ and asymptotic behaviour kicks in. We will add these details and clarify further in the main text.

[Meta-Review · NeurIPS 2020]

Strength • Paper is generally clearly written • A new and technically sound method is proposed • It provides theoretical results • Experiments are conducted Weakness • Experiments can be further improved.